# The Pivotal Role of Microscopy in Unravelling the Nature of Microbial Deterioration of Waterlogged Wood: A Review

**Adya P. Singh [1], Jong Sik Kim [2], Ralf Möller [3], Ramesh R. Chavan [4] and Yoon Soo Kim [2],***

[1] Scion (New Zealand Forest Research Institute), Rotorua 3046, New Zealand; adyasingh@hotmail.com
[2] Department of Wood Science and Engineering, Chonnam National University, Gwangju 61186, Republic of Korea; jongsik.kim@jnu.ac.kr
[3] Wolman Wood and Fire Protection GmbH, 76547 Sinzheim, Germany; ralf.moeller@wolman.de
[4] School of Biological Sciences, University of Auckland, Auckland 1442, New Zealand; r.chavan@auckland.ac.nz
* Correspondence: kimys@jnu.ac.kr

**Abstract:** This review focuses on the pivotal role microscopy has played in diagnosing the type(s) of microbial attacks present in waterlogged ancient wooden objects, and to understand the nature and extent of deterioration of such objects. The microscopic journey began with the application of light microscopy (LM) to examine the deterioration of waterlogged woods, notably foundation piles supporting historic buildings, progressing into the use of high-resolution imaging tools (SEM and TEM) and techniques. Although bacteria were implicated in the deterioration of foundation piles, confirmation that bacteria can indeed degrade wood in its native state came when decaying wood from natural environments was examined using electron microscopy, particularly TEM, which enabled bacterial association with cell wall regions undergoing degradation to be clearly resolved. The information base has been a catalyst, stimulating numerous studies in the past three decades or so to understand the nature of microbial degradation of waterlogged archaeological wood more precisely, combining LM, SEM, and TEM with high-resolution chemical analytical methods, including chemical microscopy. The emerging information is aiding targeted developments towards a more effective conservation of ancient wooden objects as they begin to be uncovered from burial and waterlogging environments.

**Keywords:** microscopy; LM; SEM; TEM; waterlogged wood; cell wall degradation; soft rot; bacterial erosion; bacterial tunnelling





## 1. Introduction

Wood as a construction material has served humans for thousands of years and was even used by human ancestors who roamed the forests in search of suitable locations for food and shelter. One of the best examples of the cooperative efforts made, and the knowledge and skills used by early humans in constructing fit-for-purpose objects from wood for their use stems from the discovery of wooden hunting spears from excavations led by Hartmut Thieme between 1994 and 1998 in the Schöningen region of Germany [1,2]. The spears were uncovered from an anoxic burial site (buried 10 m deep), and the excellent preservation of the spears permitted their age to be estimated (300,000–400,000 years). They are regarded as the oldest complete wooden spears discovered. However, the half-a-million-year-old 'Lincoln Logs' recently unearthed in Zambia are even more ancient wooden structures [3]. In addition to visual assessments, the spears were examined using microscopy to identify the wood species and to investigate whether the spears displayed any symptoms of microbial decay.

Although not as ancient as the Schöningen spears, there are other examples of relatively well-preserved wooden objects and constructions after hundreds and even thousands of years of exposure to burial environments supporting waterlogging, which are reviewed

in refs. [4–6]. Among these, most extensively investigated are the wooden foundation piles supporting buildings and sunken ships [6–10], because of their cultural and historical relevance as well as for the purposes of their repair/replacement (deteriorating foundation piles) and conservation (excavated ships and shipwrecks). The most familiar examples of excavated heritage wooden ships and their wrecks examined to investigate the cause of their deterioration are the Oseberg ship from the Viking period (Norway), the Swedish warship Vasa, King Henry VIII's warship Mary Rose (UK), the German ship Cog, the 3000-year-old Uluburun ship (uncovered on the Turkish coast), and several Chinese ships, such as Nanhai No. 1.

The biodegradation of wood is a natural process which is important for carbon cycling. Wood degradation by microorganisms has been studied using a wide range of imaging and analytical tools and techniques, among which microscopy has played a pivotal role in diagnosing the type(s) of microbial decay present in wood tissues and in understanding the intricacies of the decay process in natural environments as well as under laboratory conditions. It has been confirmed that wood can be degraded by both fungi and bacteria [11–13], with white-rot and brown-rot fungi causing considerably more rapid deterioration than soft-rot fungi and bacteria [14]. Under waterlogging conditions, which create an environment where rapid degradation is hindered, white- and brown-rot fungi struggle, as they require a high level of oxygen for growth and activity. Consequently, wood is decomposed by soft-rot fungi and tunnelling and erosion bacteria, but at a significantly slower rate, as reviewed in Refs. [6,15–17]. It is therefore not surprising that wooden constructions and objects of cultural and historical importance uncovered from buried and waterlogged environments have been reported to have survived for hundreds and even thousands of years. However, the condition of their preservation varies from near-perfect state, such as the Shöningen spears [1,18], to extensive surface degradation, depending upon waterlogging/burial environments, the time of exposure, wood type, the type of microbial decay present, and the speed at which the cell wall was degraded [4,5].

Preserved ancient wooden constructions provide information about the past climate and human history and civilization. Therefore, efforts at uncovering and characterizing waterlogged heritage wooden constructions and artefacts to understand the cause and extent of their deterioration, with the aim to ultimately preserve them, have accelerated in recent years. Among the techniques and tools used, microscopy has played an important role as it has greatly facilitated the understanding of the biological cause and the extent of cell wall degradation. The information already available on specific microbial decay patterns from the examination of wood samples obtained from wood products placed in service as well as exposed to natural environments, as reviewed in refs. [11,19,20], has served as an important baseline. While it has been possible to diagnose the type of fungal decay using light microscopy (LM), based on the appearance of characteristic micromorphological patterns produced [11,13], the use of scanning electron microscopy (SEM) and transmission electron microscopy (TEM) has resulted in the capture of images at much greater resolution, revealing decay micromorphologies in greater detail, facilitating the interpretation of the mechanism of cell wall degradation [20,21]. TEM has been particularly useful in obtaining distortion-free high-resolution images with greatly enhanced definition, importantly aiding the understanding of the nature of wood cell wall degradation (i.e., the intricacies of the pattern of cell wall degradation). The images acquired using TEM have thus confirmed that bacteria can degrade the wood cell wall in its native state, as reviewed in Refs. [14,19,22], although the presence of bacteria in decaying wood has been recognized for a long time, as reviewed in Refs. [23,24]. The information obtained proved to be vitally important for diagnosing the type(s) of microbial decay present in waterlogged woods and has spurred a flurry of activities in recent years related to understanding the deterioration and conservation of culturally and historically important and much treasured wooden constructions and artefacts, such as the Schöningen spears [1,25] and ships uncovered from waterlogged burial sites supporting an anoxic environment. Using electron microscopy, it has been revealed that in oxygen-depleted environments, wood is primarily degraded by erosion

bacteria [26–28], and that cell wall degradation is extremely slow, allowing waterlogged wooden artefacts to survive centuries of exposure to anoxic burial environments [4,5].

For waterlogged woods, recent microscopic studies have provided evidence of cell wall degradation by cavity-producing soft-rot fungi (Type I), tunnelling bacteria, and erosion bacteria, or predominantly by erosion bacteria [6,9,15–17,27,29–31]. Cell wall erosion-type attacks by soft-rot fungi (Type II) were rarely observed in such woods [31]. Attacks by soft-rot and tunnelling bacteria are present in waterlogged wood exposed to oxygenated environments, such as intertidal zones [9,30,32], which suggests that these organisms require oxygen for their activity. Erosion bacteria are the main degraders of waterlogged wood under anoxic conditions [7,16,17,27,28]. The effective conservation of buried and waterlogged wooden objects relies on detailed and accurate information from studies involving integrated multidisciplinary approaches on wood type, anatomical characteristics, decay grading, the physical and chemical state of wood, ash content, and the types of inorganic elements present [33–36]. Furthermore, it is important to have an understanding of the micromorphological and chemical characteristics of the outer heavily degraded wood tissues and those in transition between the outer degraded and the inner unaffected wood tissues. It is important to have a knowledge of the organization of wood at both tissue and cell wall levels, as well as familiarity with the patterns of decay produced by the microorganisms known to be associated with the cell wall degradation of waterlogged woods. This will help to determine the cause and extent of the deterioration of historical and cultural wood properties exposed to a particular environment so that an assessment can be made as to how best to preserve such a precious resource. Therefore, this review will briefly cover wood structure and composition, the micromorphological patterns associated with the above three types of wood decay microorganisms, and the role of microscopy in understanding the microbial degradation of cell wall in relation to cell and cell wall features, and in diagnosing the type(s) of microbial decay present in waterlogged woods. The concluding remarks emphasize the opportunities present based on the advances made in microscopic and chemical characterization to develop technologies to improve the quality of preservation of ancient waterlogged wooden objects as they continue to be uncovered.

## 2. Composition and Structure of Wood

Uncovered buried and waterlogged ancient wooden objects vary in wood species that differ in tissue types and cell wall composition, as well as in the type of microbial decay present. Wood type, cell wall ultrastructure, the composition and distribution of cell wall components, and waterlogging environments all influence the type of microbial decay and the speed and extent of cell wall degradation. Therefore, knowledge of wood structure and composition and the nature and extent of cell wall degradation is important for the proper conservation of archaeological finds. However, volumes have been written, and the composition and structure will only briefly be described here.

### 2.1. Composition

The main components of wood are polymeric substances called cellulose, hemicellulose, and lignin, which make up the fabric of the cell wall. These polymers differ in their composition and also vary in their proportion, depending upon wood and cell type and across the cell wall (Table 1).

Cellulose is the most dominant component of wood cell walls and is responsible for much of the mechanical strength of the cell wall. Hemicelluloses form a link between cellulosic microfibrils and regulate aggregation of microfibrils. Lignin, a complex phenolic polymer, forms a link with the hemicellulose. It provides compressive strength and rigidity. Among the three major cell wall polymers, lignin is the most recalcitrant component, providing resistance to cell walls against attack by most wood-degrading microorganisms, and thus protecting the carbohydrate components of the cell wall [6]. Lignin is the most important component of the cell wall from the perspective of the long life of buried and waterlogged ancient wooden structures, as it provides considerable resistance to wood-

degrading erosion bacteria which, by virtue of their high tolerance to oxygen-limiting conditions, are the main wood degraders present in waterlogged environments [26,28]. Extractives are nonstructural components but can provide considerable resistance to wood-degrading microorganisms, particularly the phenolic extractives. Heartwoods are rich in extractives, and therefore their presence is relevant to the longevity of woods exposed to wet buried environments.

**Table 1.** Chemical compositions of softwood and hardwood as a common pulpwood species.

| Chemical Composition (%) | Softwood (SW) | | Hardwood (HW) | |
|---|---|---|---|---|
| | Normal Wood | Compression Wood | Normal Wood | Tension Wood |
| Cellulose | 37–43 | 29–31 | 39–45 | 50–65 |
| Galactoglucomannan (SW) Glucomannan (HW) | 15–20 | 9–12 | 2–5 | 2–4 |
| Arabinoglucuronoxylan (SW) Glucuronoxylan (HW) | 5–10 | 6–8 | 15–30 | 16–23 |
| Galactan | | 9–11 | | 0–10 |
| Lignin | 25–33 | 37–40 | 20–25 | 16–20 |
| Extractives | 2–5 | 2–5 | 2–4 | 2–4 |

The data are reproduced from Sjöström, E.; Westermark, U. Chemical composition of wood and pulps: basic constituents and their distribution. In *Analytical Methods in Wood Chemistry, Pulping, and Papermaking*; Sjöström, E., Alén, R., Eds.; Springer, Berlin, Germany, 1999; p. 3 [37].

The above description of the composition pertains to normal wood. However, there are woods that compositionally differ from normal wood and have been reported to be present in buried and waterlogged structures, such as compression wood. The presence of compression wood is very relevant to the longevity of waterlogged archaeological woods because of its considerable resistance to wood-degrading bacteria [25,38,39], particularly to erosion bacteria [6,25,27,28]. Compositionally, compression wood contains less cellulose and a greater amount of lignin, and compression wood cell walls contain a significant amount of the hemicellulose galactan [40]. The excellent preservation of the ancient wooden Schöningen spears after 400,000 years of exposure in a buried environment is an excellent example of the important role compression wood plays in protecting wood exposed to anoxic waterlogging environments, from which many important ancient wooden treasures have been uncovered, such as that documented through the electron microscopic examination of wood from the Schöningen spears, showing the presence of mild compression wood [25], which is known to resist bacterial erosion [38,41].

### 2.2. Structure

Softwoods are predominantly composed of one cell type (tracheids) in their axial system. Hardwoods are composed of several types of cells, namely, vessel elements, fibres, fibre-tracheids, and parenchyma. Generally, the radial system (rays) consists of two cell types, parenchyma and tracheids. Softwood rays contain parenchyma and some in addition contain tracheids (radial tracheids), whereas hard wood rays consist only of parenchyma cells [42].

After a few years of growth, trees begin to infiltrate the core wood with extractives (secondary metabolites). This part of the wood is referred to as heartwood and the process is called heartwood formation [43]. Heartwood is not only functionally important for growing trees, but it is also relevant to the long life of buried and waterlogged archaeological woods, because the presence of abundant extractives provides such woods with resistance to degradation by microorganisms, which are the main factors for the deterioration of wood exposed to anoxic waterlogged environments.

From the perspective of microbial invasion into waterlogged woods, ray parenchyma, being rich in nutrients, serves as the initial point of microbial entry into wood. Rays are connected to the elements of the axial system. The spread of microorganisms from rays into axial elements and subsequently among axial elements occurs most readily via pits, as the absence of lignin in pit membranes enables cellulolytic microorganisms to readily degrade the membrane [44]. Other wood structures/features that are relevant from the perspective of the microbial deterioration of waterlogged woods, such as compression wood, warts, vestures, high lignin containing cell wall structures, and regions, will be covered in a separate section.

*2.3. Cell Wall Ultrastructure*

The organization of cell walls at the ultrastructural level is relevant to understanding the nature and extent of the microbial deterioration of waterlogged wood, as the mode of attack by wood-degrading microorganisms (soft-rot fungi, tunnelling bacteria, and erosion bacteria) reported to be present in buried and waterlogged woods varies. The cell wall organization is described here for lignified wood elements (tracheids, vessel elements, and fibres) which resist degradation and are thus highly relevant to the long-term survival of wooden structures exposed to burial and waterlogged environments.

The ultrastructural organization of the cell wall of lignified wood elements is similar. The cell walls consist of a middle lamella, a primary wall, and a secondary wall. The middle lamella is the first structure to develop post-mitosis between the products of cambial cell division and at the time of cytokinesis. The middle lamella is rich in pectic polysaccharides. As mentioned in the sub-section 'Composition', the secondary wall is composed of three main polymer types: cellulose, hemicellulose, and lignin. The cellulose forms as long chains of glucose units at the outer face of the cell membrane (plasma membrane) from the activity of cellulose synthase complexes, as reviewed in ref. [45], which, when viewed in freeze-fracture preparations using TEM, display a distinctive form, referred to as the 'rosette' [46,47]. Cellulose consists of linear chains of $\beta$-(1-4) glucans, and aggregated parallel chains are stabilized by hydrogen bonding, culminating into a basic structural unit referred to as the microfibril. The orientation of cellulose deposition in the cell wall is regulated by the microtubules present in the outer region of the cytoplasm underlying the plasma membrane (cortical microtubules) [48]. The orientation of microfibrils in the wood cell wall is a major determinant of strength and stiffness properties of wood [49] and is also relevant from the perspective of the microbial deterioration of buried and waterlogged wooden objects, as we will see later. Hemicelluloses xylan (common to angiosperm woods) and mannan (common to gymnosperm woods) are synthesized in the Golgi and packaged into vesicles. The vesicles containing hemicelluloses are released (budded off) from Golgi cisternae and transported to the plasma membrane, as reviewed in Ref. [50], where contents are released into the developing cell wall upon fusion of vesicles with the plasma membrane. The delivered hemicelluloses form linkages with microfibrils. The microfibril–hemicellulose linkages have been visualized as bridges/cross-links in both primary and secondary cell walls using a range of high-resolution microscopic techniques [51–53]. It is assumed that the time and space of hemicellulose delivery is closely coordinated with cellulose formation for precise cell wall architecture to develop [50]; however, this aspect of cell wall formation has not been well documented.

Lignin is incorporated into the cell wall subsequent to the establishment of the microfibril–hemicellulose architecture, initially as monolignols (*p*-coumaryl alcohol, coniferyl alcohol, and sinapyl alcohol) produced in the cytosol. Lignin polymerization occurs when the monolignols are oxidized to produce monolignol radicals. The understanding of how monolignols are delivered to the developing cell wall is incomplete. It is widely accepted that peroxidases play a role in the oxidation of monolignols, and may determine the site of initial lignification, the cell corner middle lamella, from where lignification progresses into the forming secondary cell wall [54]. The polymerizations of the above types of monolignols gives rise to lignin types referred to as S, G, and H Lignins, the presence and

amount of which vary with wood and cell types. Lignin, a complex aromatic polymer, is the most recalcitrant component of the cell wall. The type, concentration, and location of lignin in the cell wall are important from the perspective of the microbial and enzymatic deconstruction of the cell wall, and therefore relevant to the microbial deterioration of buried and waterlogged wood. Because wood is an important natural renewable resource for biofuel production, the knowledge of lignin type and content is also important for developing bio-refineries that can yield cost-effective biofuels [55].

Although macro, micro, and nano levels of organization of wood all are important from the perspective of wood biodeterioration, it is the cell wall ultrastructure (molecular and structural aspects) that is most relevant to understanding cell wall degradation by microorganisms known to deteriorate wood buried and waterlogged for prolonged periods. The ultrastructural organization of conifer tracheids has been most extensively investigated, but it is also common to the lignified elements of hardwoods. The secondary wall, being the thickest component of tracheids, is the main source of nutrients for the microorganisms (soft-rot fungi, tunnelling bacteria, and erosion bacteria) attacking wood exposed to waterlogging conditions [5]. These organisms differ not only in the mode of their action but also in their capability to attack compositionally and structurally different regions of the cell wall. Although identifiable at the light microscope (LM) level, the resolution of the electron microscope is much greater, revealing the fine structure of the cell wall in considerable detail. A range of microscopy tools and techniques alone and in combination, such as LM combined with histochemistry, fluorescence confocal laser microscopy, Raman microscopy and spectroscopy, UV microscopy, FTIR microscopy, conventional and field-emission SEMs, and TEM in combination with potassium permanganate ($KMnO_4$) staining and immunolabelling of ultrathin sections, have been applied to unravel the intricacies of cell wall organization, as reviewed in Ref. [42].

### 2.3.1. The Typical Three-Layered Secondary Wall

The secondary wall is laid down subsequent to the formation of the middle lamella and the primary wall. The secondary wall is a three-layered structure (Figure 1), the layers of which are referred to as S1 (S referring to the secondary wall), S2, and S3, based on the sequence in which they are deposited. The outermost (S1) and the innermost (S3) secondary wall layers are considerably thinner than the middle layer (S2), which is most prominent in the latewood of gymnosperms and the fibres of angiosperms. In normal wood, the three layers vary in the orientation of cellulose microfibrils and, to a degree, also in chemical composition [42,49,56,57]. The microfibrils in the S2 layer are oriented in the long direction of wood elements, the angle of which varies with cell type and wood species. The S1 and S2 layers display an orientation essentially perpendicular to the microfibril orientation in the S2 layer, a cell wall design that maximizes the mechanical properties of wood (Figure 2). Although not illustrated in many published secondary cell wall models, including a recent publication [56], a transition in the microfibril orientation at the interface of S1-S2 and S2-S3 with a gradual shift has been noted, particularly using TEM [58], a design apparently evolved to minimize interfacial delamination under stresses generated during growth and imposed by external factors. While the S2 layer, as the thickest secondary wall layer, makes the largest contribution to the mechanical properties of wood, the very thin S1 and S3 layers make their own specific contributions which are important for the structural integrity of wood cell walls. The S1 layer prevents the swelling of the cell wall by virtue of its microfibril orientation perpendicular to the long direction of wood cells. The S3 layer contributes greatly to the resistance of the cell wall, preventing it from collapsing into the lumen [59–61] under tensile stresses generated during water transport in living trees and from other pressures, such as compression parallel to the wood grain. Cellulose is not only the most dominant component of wood cell walls in terms of quantity and its contribution to the mechanical properties of wood, but the orientation of microfibrils in the cell wall is highly relevant to the microbial deterioration of waterlogged woods attacked mainly by

erosion bacteria, as it plays a decisive role in regulating the direction of movement of these bacteria [15,28].

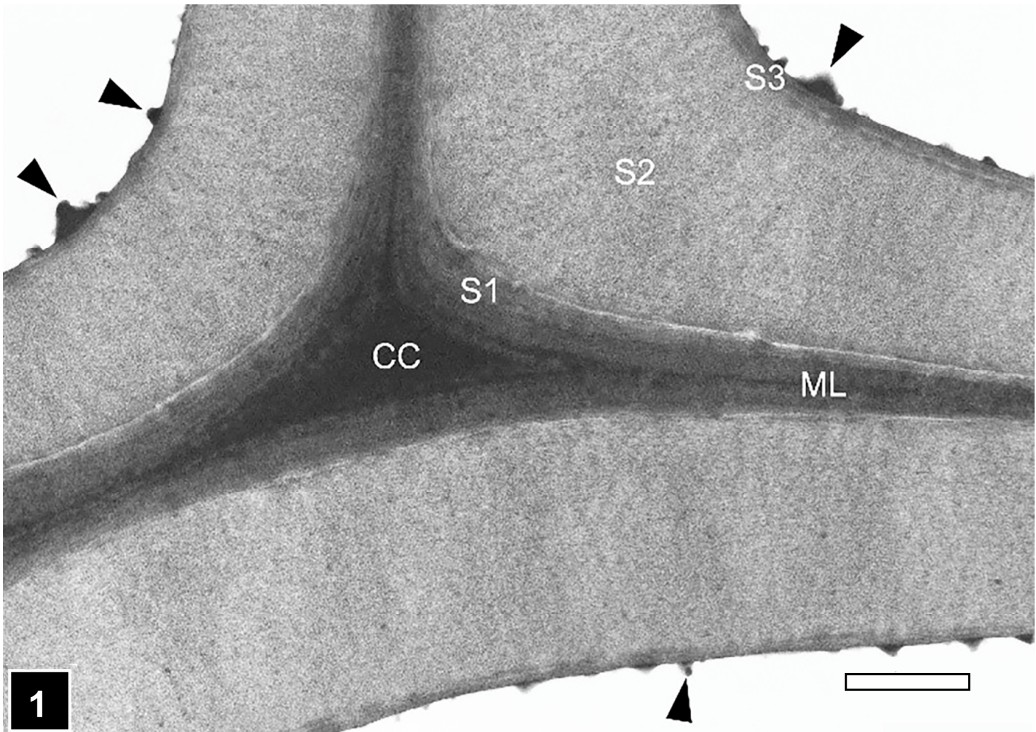

**Figure 1.** TEM of a transverse section through a cell corner region between tracheids of silver fir. The secondary cell wall is a three-layered structure. CC, cell corner; ML, middle lamella; S1, S2, S3, secondary wall layers; warts (arrowheads). Scale bar = 1 μm. The image is reproduced from Singh et al. [6].

From the perspective of the mechanical performance of wood, while cellulose is a key player in determining the strength and stiffness of wood cell walls; it is the interaction of the three major cell wall components, cellulose, hemicellulose, and lignin, that has an overall effect on the mechanical behaviour of wood. Hemicelluloses form associations with cellulose as well as lignin, and thus play a role in the establishment of an interconnecting system both in physical and functional terms. Hemicelluloses play a role as structural regulators, preventing the bundling (aggregation) of cellulose microfibrils by establishing direct linkages with microfibrils [62]. Berglund et al. [63], deploying an experimental system, demonstrated that the hemicelluloses xylans and mannans make distinct biomechanical contributions to the integrity of secondary cell walls in tension as well as compression. Their association with cellulose and molecular structure jointly play a role in this function. Hemicelluloses, unlike cellulose, are complex branched polymers; their molecular heterogeneity is a key factor in regulating supramolecular interactions with cellulose surfaces via hydrogen bonding and non-polar interactions [64]. Overall, a collective contribution from cellulose, hemicellulose, and lignin is responsible for all physical and mechanical properties of plant and wood tissues, namely, strength, stiffness, toughness, hardness, and rigidity [65,66].

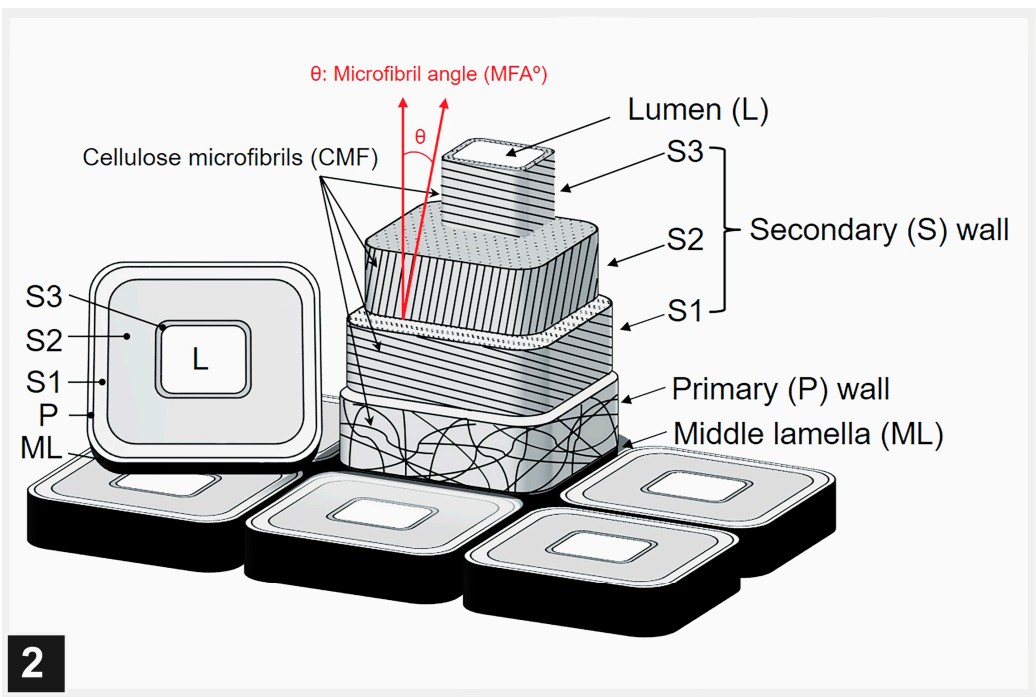

**Figure 2.** Schematic representation of cell wall model showing the layered cell wall and different orientation of cellulose microfibrils in the cell wall layers.

From the perspective of the contributions that cell wall polymers make to the resistance of wooden objects to biodegradation in the anoxic burial environment resulting from waterlogging, lignin serves as the most important factor. This is because lignin, a polymeric structure that results from the polymerization of different monolignols through oxidative coupling, is highly irregular (inhomogeneous), and erosion bacteria may not have evolved the mechanism to completely degrade this polymer and have only the capability of uncoupling it from hemicelluloses to gain access to cell wall carbohydrates. Lignification begins in the cell corner middle lamella and progresses to other regions of the cell wall (inter-corner middle lamella and secondary cell wall) from this site [54,67]. In normal wood, the cell corner middle lamella region is also the most highly lignified part of the wood cell wall, with the inter-corner middle lamella being the second most highly lignified part. These, together with the primary cell wall, which is also highly lignified, are the only cell wall regions that are retained in buried and waterlogged wood tissues heavily degraded by erosion bacteria.

From the perspective of cell wall resistance to the microbial degradation of waterlogged woods, the S3 layer is also important, as it is directly exposed to the cell lumen which the microorganisms colonize and from where they have to breach the S3 layer to gain access to the nutrient-rich S2 layer. In conifers, where the lignin concentration of the S3 layer is significantly high, such as the *Pinus radiata* S3 layer with over 50% lignin [68], this layer also exhibits considerable resistance to degradation by erosion bacteria. In the TEM images of erosion bacteria-degraded conifer wood tissues, relatively large segments of the S3 layer can often be seen to persist even after the extensive degradation of the S2 layer in conifer woods exposed to near-anaerobic environments [26], including waterlogged archaeological woods [9]. This points to the evolution of a clever strategy by erosion bacteria to degrade only a part of the lignin-rich S3 layer sufficiently large for them to traverse it to access the S2 layer, as the lignin degradation is considered to be an energy-demanding process. The resistance of the S3 layer to erosion bacteria in such situations no doubt affects the speed of bacterial degradation, which helps prolong the life of wooden objects exposed to waterlogged environments. The S3 layer in softwood and some hardwood species is often covered with warts and a warty layer (Figure 1), another protective factor in relation to the

survival of waterlogged wooden objects, as this layer, together with the S3 layer, is a major barrier for the wood-degrading microorganisms attempting to gain access to the S2 wall.

2.3.2. Multilamellar Cell Walls

In some plants, fibre cell walls display a design incorporating the presence of more than three layers, where alternating layers differ in the orientation of cellulose microfibrils. This type of cell wall design is considered to enhance mechanical properties of wood, much like the contribution of cross-lamination to the high strength and stiffness of laminated timber panel products [69]. Where associated with thin-walled tissues, such as parenchyma and phloem tissues, fibres have a mechanical support function [70]. The multilamellar secondary cell wall of fibres consists of more than four lamellae. The cell wall is differentiated into alternating thick (wide) and thin (narrow) lamellae, and this arrangement is consistent across the cell wall, although variations in the thickness have been encountered, particularly for the thick lamellae. The thick and thin lamellae differ in the orientation of cellulose microfibrils, with the thick lamellae displaying an orientation nearly parallel to the fibre axis and the thin lamellae nearly perpendicular to the fibre axis [71–74].

Compositionally, there are indications that the thin lamellae of the multilamellar fibre walls are more highly lignified than the thick lamellae, based primarily on the assessment made from TEM observations of ultrathin sections of wood after $KMnO_4$ staining and by means of UV microscopy and spectroscopy at 280 nm absorption specific for lignin [73]. Support also comes from the analysis of TEM images displaying an unusual form of soft rot cavities. The usual form of soft rot cavities, as observable in transverse sections, is circular or near-circular (Figures 3 and 4). However, the cavities interfacing thin lamellae in multilamellar fibre cell walls appear to be crescent-shaped (half-moon-shaped) cavities being flattened along the thin lamellae [73,75,76] (Figure 4), which suggests an influence of lignin concentration on the development of soft rot cavities.

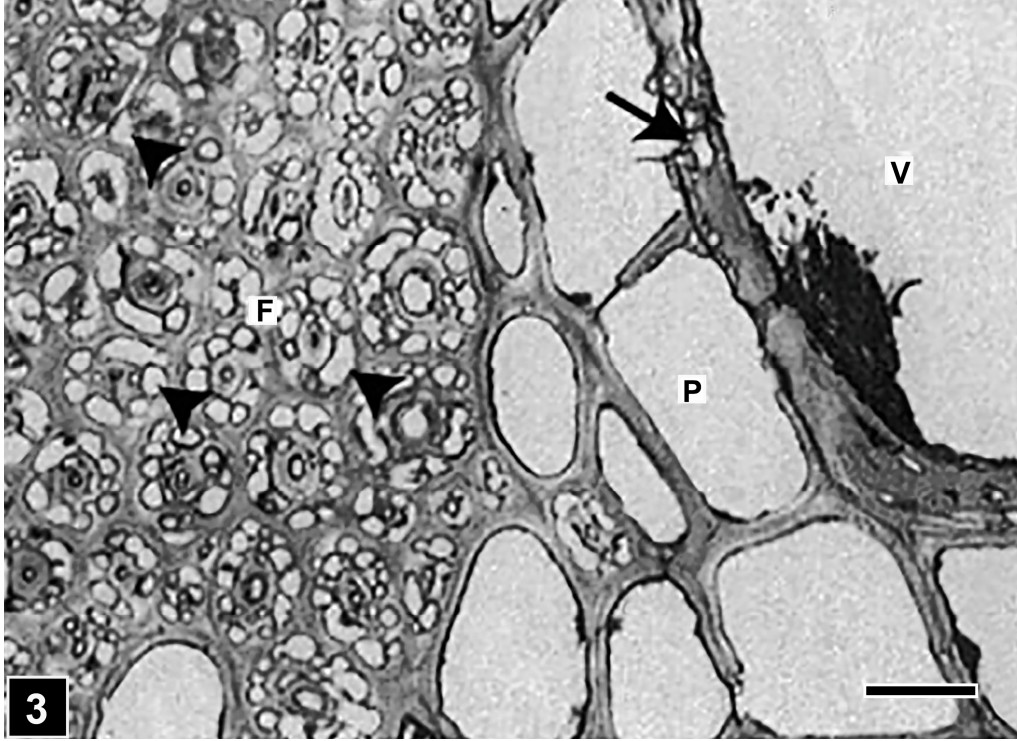

**Figure 3.** LM of a transverse section through a hardwood tissue attacked by soft rot I. Cavities are abundantly present in fibre cell (F) walls (arrowheads) but are few in the vessel (V) wall (arrow). Individual (non-coalesced) cavities appear to be circular-shaped. P, parenchyma. Scale bar = 20 μm. The image is reproduced from Singh et al. [75].

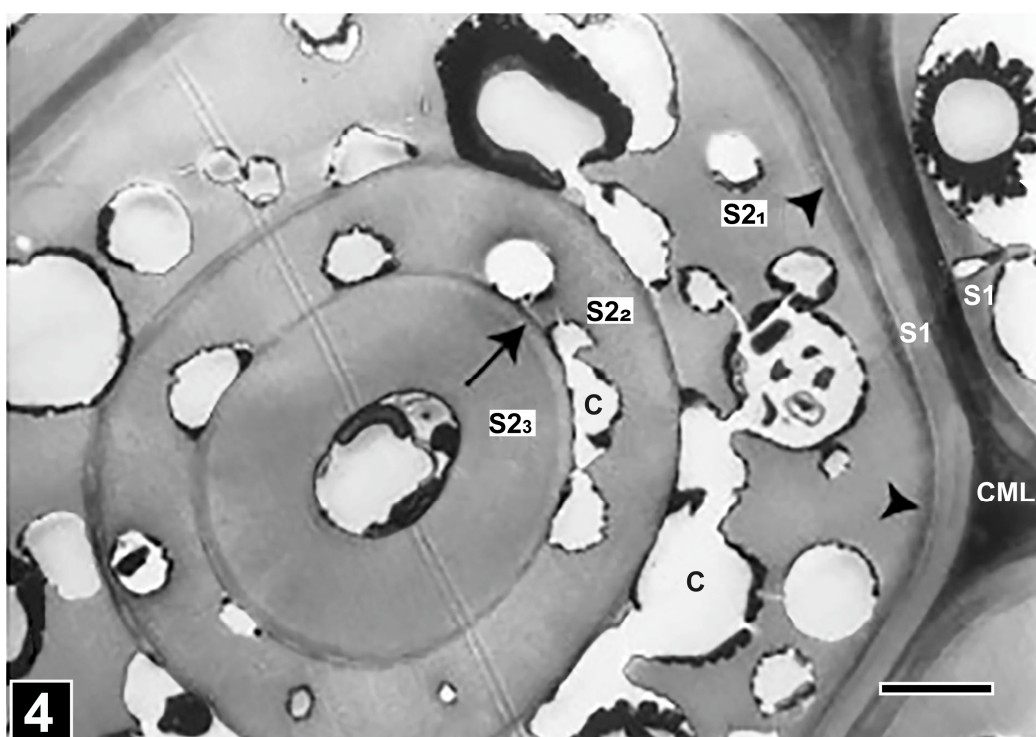

**Figure 4.** TEM of a transverse section through a multilamellar fibre cell wall, composed of thick lamellae (S2-1, S2-2, S2-3) alternating with very thin lamellae (arrow). The face of cavities in contact with the thin lamellae (C) has a flattened appearance. The arrowheads indicate a highly lignified part of the S2-1 underlying the S1 wall. CML, compound middle lamella. Scale bar = 2 μm. The image is reproduced from Singh et al. [75].

### 2.3.3. Compression Wood Cell Walls

Gymnosperm trees have evolved a mechanism for up-righting their leaning stems and branches by forming compression wood, a type of reaction wood, which develops on the underside of such stems and branches in response to gravity [40,77]. The presence of compression wood in waterlogged wooden objects is highly relevant in regard to its positive contribution to the longevity of such objects in long-term exposure to burial and/or water submerged environments. Therefore, understanding the structure and composition of compression wood is important in terms of the cell wall features responsible for the resistance to the wood-degrading organisms reported to be present in waterlogged woods, particularly to erosion bacteria, which are ubiquitous in woods exposed to anoxic waterlogged environments. Chemically, compared to normal wood, all forms of compression wood contain a high proportion of lignin and less cellulose, a feature most pronounced in severe compression wood. Another important feature of compression wood is the presence of high amounts of galactan.

The compression wood form varies greatly. Based on the microscopic features, compression wood has been reported to range from very mild forms, which exhibit a near-normal appearance of cells, to severe forms, exhibiting cell wall features that differ greatly from those of normal wood cells [40,49,78,79]. It is not uncommon to observe a gradation in compression wood forms ranging from mild to severe compression wood within the same disc of a stem or branch, particularly in fast-growing trees, such as the radiata pine in New Zealand. The very mild forms can readily be missed if specialized microscopic techniques for assessing lignin concentration in the cell wall, such as UV microscopy, confocal fluorescence microscopy, and TEM in combination with the $KMnO_4$ staining of ultrathin sections [25,78–80], are not employed. At the cell wall level, such forms are characterized by the presence of a thin cell wall, the absence of intercellular spaces, the increased lignification of the outer S2 wall ($S2_L$ layer) confined to the cell corner region, and the reduced

lignification of the middle lamella compared to normal wood cells [79]. The S3 layer may or may not be present [79,81]. With an increase in severity, some of the features characteristic of severe compression wood, such as the rounding of cells, the presence of intercellular spaces in cell corners, the lack of the S3 layer, and an increase in lignification of the $S2_L$ layer continuing around the perimeter of the cell [78], are also present in mild and moderate compression wood forms, pointing to a transition into severe compression wood. However, these features are considerably more pronounced in severe compression wood, with a distinct rounding of cells, intercellular spaces in cell corners, an increased lignification of the $S2_L$ layer extending to occupy a large proportion of the S2 wall, helical cavities in the S2 wall, and the absence of the S3 layer [40,77]. Among compression wood forms, the presence of a higher amount of lignin and less cellulose compared to normal wood is most pronounced for severe compression wood. The presence of a significant amount of galactan in the cell wall is one of the most characteristic features of compression wood and is regarded as the biological marker for this type of wood [82]. Immunolocalization microscopic techniques have demonstrated that the location of galactan in the cell wall corresponds to the most highly lignified region of the S2 wall [82–85].

The presence of compression wood can no doubt make significant contributions to the longevity of wooden objects exposed to waterlogged environments over prolonged periods. In this respect, the lignin-enrichment of the $S2_L$ layer is the most important feature, as it enhances the resistance of the cell wall to microorganisms degrading waterlogged woods, particularly to erosion bacteria. This applies to both severe and mild compression wood, although the resistance increases with the severity of compression wood, specifically the concentration and distribution of lignin in the $S2_L$ layer. Kim and Singh [29], examining wood samples from several ancient sunken ships from different sites and burial and waterlogging conditions, provided evidence of the almost complete resistance of the highly lignified $S2_L$ layer in severe compression wood cells to erosion bacteria. There is also some information available on the bacterial degradation of cell walls in mild compression wood. The TEM examination of mild compression wood in timbers placed in service in an industrial cooling tower provided an opportunity to compare the degradation of mild compression wood cell walls by tunnelling and erosion bacteria in the same as well as neighbouring tracheids [38,86]. While tunnelling bacteria were able to penetrate deep into the $S2_L$ in places, judging by the presence of the characteristic tunnels they form, bacterial erosion was restricted to the innermost regions of the $S2_L$ layer. Perhaps the most outstanding example is the near-perfect preservation of the 400,000-year-old Schöningen wooden spears uncovered from an anoxic burial environment, and which contained compression wood [25].

### 2.3.4. Tension Wood Cell Walls

Tension wood is a type of reaction wood which develops on the upper side of leaning branches and stems in angiosperm trees, a mechanism that trees have evolved for up-righting the bent branches and stems [87]. Compared to normal wood fibres, tension wood fibres contain more cellulose and less lignin. Tension wood has been estimated to contain 5 to 10% less lignin [88] and 10% or more cellulose [89] compared to normal wood. Fibres develop a special layer in the innermost part of the cell wall (facing the cell lumen), replacing the S3 layer, which is referred to as the gelatinous layer (G-layer). The G-layer is mainly cellulosic in composition but may also contain small amounts of pectins, hemicelluloses, and arabinogalactan proteins, as reviewed in Ref. [87]. Traces of lignin and other phenolic substances have also been detected in the G-layer [90–92]. Cellulose microfibrils in the G-layer are oriented almost parallel to the long direction of the fibre. With respect to relevance to the longevity of ancient waterlogged woods, the contribution of tension wood is expected to be insignificant, as the presence of a G-layer is considered to render tension wood fibres more susceptible to wood-destroying fungi than normal wood fibres [13]. However, in a recent study of the microbial decay in waterlogged archaeological rosewood, Kim et al. [31] found the G-layer in tension wood fibres to show

some resistance in the initial stages of attack by soft-rot fungi. From a biotechnological perspective, the unique features of tension wood fibres, particularly the presence of the cellulose-rich G-layer, are being exploited in developing processes to enhance bioethanol production from woody biomass rich in tension wood. With high amounts of cellulose and less lignin in tension wood fibres compared to normal wood fibres, tension wood is proving to be an attractive substrate for enzymatic digestion aimed at producing biofuels [93,94].

## 3. Microscopic Techniques

Microscopy has played a crucial role in understanding the microbial degradation of cell walls, and the information obtained, particularly related to the micromorphological appearance of degradation patterns, has proved vitally important in diagnosing the type(s) of decay patterns present in ancient waterlogged woods (Table 2).

Research in the past three decades or so has provided evidence of the presence of soft rot, bacterial tunnelling, and bacterial erosion in waterlogged woods. Although soft-rot decay could be readily distinguished using LM, it was not until the application of high-resolution tools, such as SEM and TEM, that it became possible to clearly resolve the unique decay patterns produced by bacteria and thus to confirm that wooden objects subjected to prolonged waterlogging can also be degraded by bacteria, as reviewed in ref. [6]. The information on the micromorphological characteristics of the above-mentioned decay types, based on the examination of decaying woods obtained from various natural environments as well as from laboratory exposure studies, has appeared in several reviews [4,6,11,15,17, 19,22,23,28,30], and therefore the decay patterns are only briefly described here, with the aim to benefit particularly those beginning to explore the understanding of the microbial deterioration of waterlogged woods.

With respect to the suitability of microscopic tools and methods for extracting information particularly relevant to the conservation of ancient waterlogged wooden objects, and in consideration of the volume of material available, it is first important to critically analyse the value that each method can offer. Conventional light microscopy (LM) is a simple method that can permit the rapid assessment of a large volume of material without the need for time-consuming and tedious processes associated with high-resolution electron microscopy, particularly TEM. The other advantage is that qualitative information on the chemical nature of cell walls undergoing degradation and residues can be obtained using histochemical methods, such as the staining of sections with toluidine blue and/or phloroglucinol-HCl (Weisner reagent). Correlated LM-TEM can be undertaken by processing pieces of sections from regions of interest for examination with TEM after LM viewing, and both stained and unstained sections can be processed. Thus, chemical information can be combined with high-resolution structural information, for instance, the texture of un-degraded cell walls and those undergoing degradation. LM-SEM correlated microscopy is also possible, the limitation being the viewing of highly degraded, fragile tissues with SEM. However, sections can be embedded in an embedding medium that can be removed prior to SEM imaging at a low kV. But it may not be possible to obtain cell wall textural information in the same detail as with TEM. Additionally, among the decay types reported for waterlogged woods, while the presence of soft rot Type I and bacterial erosion can be diagnosed using LM, it is difficult to unambiguously confirm the presence of bacterial tunnelling, despite marked improvements in the definition of cellular structures by examining semi-thin sections cut from polymer-embedded tissue blocks using an ultramicrotome.

**Table 2.** Comparison of microscopic techniques commonly used in the studies of archaeological waterlogged woods.

| Microscopy | Conventional LM | Conventional SEM | Conventional TEM | CLSM |
|---|---|---|---|---|
| Features | • Rapid assessment of a large volume of decayed wood cells.<br>• Provides qualitative chemical information on the decayed wood cell wall and residuals, combined with histochemical staining.<br>• Difficulty in confirming certain decay types, such as bacterial tunnelling.<br>• Limited resolution and difficulty in obtaining clear focus at high magnifications. | • Three-dimensional imaging of the microstructure of decayed wood cell walls.<br>• Provides information on the micro-distribution of inorganic elements in decayed wood cell walls, combined with SEM-EDS (or EDX) mapping.<br>• Large wood samples (~several cm) are available for observation.<br>• Time-consuming with sample preparation and imaging; mainly surface features but probing to certain depth possible. | - High-resolution imaging of the internal structure of decayed wood cell walls.<br>- Provides information on the micro-distribution of inorganic elements in decayed wood cell walls, with TEM-EDS (or EDX) mapping.<br>- Complex sample preparation and limited field of view, particularly compared to SEM.<br>- Difficulty obtaining a three-dimensional understanding of wood cell walls. | - Relatively higher resolution images than traditional LM.<br>- Image collection from various depths through the optical sectioning of decayed wood cell walls enables reconstruction.<br>- Very useful for analysing lignin distribution in decayed wood cell walls.<br>- Limited resolution compared to electron microscopy (i.e., SEM and TEM). |
| Common sample preparation methods | • Fixation (optional)<br>With GA or GA + PA<br>Softening (optional)<br>  - Hard materials only (e.g., dense tropical hardwood) using a mixture of water and glycerine.<br>• Embedding (optional)<br>  - Soft materials only (highly degraded) using paraffin and various resins (e.g., LR-white, Spurr, and Epon).<br>• Sectioning<br>  - Hand-section using a razor blade.<br>  - Microtome-section both from embedded (typically 0.5–1 μm thickness) and non-embedded samples (typically 10–30 μm thickness). | • Fixation<br>  - With GA or GA + PA.<br>  - Post-fixation with $OsO_4$ (optional).<br>• Drying<br>  - Critical point-drying using liquid $CO_2$.<br>  - Freeze-drying, combined with *t*-butyl alcohol.<br>• Embedding (optional)<br>  - Very fragile materials only (highly degraded) using paraffin and various resins (e.g., LR-white, Spurr, and Epon).<br>  - Embedding media are typically removed prior to observation.<br>• Sectioning (optional) using microtome (various from μm to several cm thickness) | • Fixation<br>  - With GA or GA + PA;<br>  - Physical fixation: rapid freezing;<br>  - Post-fixation in $OsO_4$ (optional);<br>  - Post-fixation with $OsO_4$ is omitted for immunocytochemistry.<br>• Resin embedding in acrylic (e.g., LR-white) and epoxy resins (e.g., Spurr and Epon)<br>  - Acrylic resins are preferred than epoxy resins for TEM-immuno-cytochemistry.<br>• Sectioning using ultramicrotome equipped with diamond knife (typically 70–100 nm thickness) | • Fixation<br>  - With GA or GA + PA<br>• Embedding<br>  - Resin embedding in acrylic (e.g., LR-white) and epoxy resin (e.g., Spurr, Epon).<br>  - Acrylic resins are preferred to epoxy resins for immuno-fluorescence labelling.<br>• Sectioning using microtome (typically 0.5–1 μm thickness).<br>• Mounting sections in glycerol or anti-fade fluorescence agents (optional). |

**Table 2.** *Cont.*

| Microscopy | Conventional LM | Conventional SEM | Conventional TEM | CLSM |
|---|---|---|---|---|
| Staining or coating | • Both stained and un-stained sections can be processed.<br>• Histochemical staining using a dye such as toluidine blue, safranin, and phloroglucinol-HCl. | • Coating with a thin layer of metals such as gold (Au) and platinum (Pt). | • Staining using lead citrate and/or uranyl acetate.<br>• Potassium permanganate ($KMnO_4$) staining. | • Both stained and un-stained (using auto-fluorescence) sections can be processed.<br>• Staining using fluorochromes such as acriflavine and acridine orange. |
| Advanced applications | • Cryo-sectioning using cryo-microtome for the near-native state of decayed wood cell walls.<br>• Observations combined with other types of optical LM, such as UV microscopy.<br>• Polarization microscopy useful to characterize bacterial eroded cell walls. | • High-resolution FE-SEM provides greater detailed decayed wood cell walls than conventional SEM.<br>• Environmental-SEM allows observations of decayed wood cell wall under moist conditions.<br>• Cryo-(FE)-SEM for the near-native state of decayed wood cell walls. | • High-resolution FE-TEM provides greater details of decayed wood cell walls than conventional TEM.<br>• TEM-tomography allows for the 3D imaging of decayed wood cell walls.<br>• Cryo-TEM for the near-native state of decayed wood cell walls.<br>• TEM-immunogold labelling using antibodies allows chemical information combined with high-resolution structural information. | • Three-dimensional reconstruction of decayed wood cell wall.<br>• Immunofluorescence labelling using antibodies allows chemical information combined with structural information. |

GA: glutaraldehyde; PA: paraformaldehyde; FE: field emission.

Confocal laser scanning microscopy (CLSM) is a versatile technique with many applications in wood research, as reviewed in ref. [95], including measuring cell dimensions, the determination of microfibril angle, the tracing penetration of protective coatings and wood modification agents, and studying cell wall polymer composition, concentration, and distribution. Fluorescence microscopy, particularly based on CLSM, has proved to be a valuable microscopic approach in imaging plant and wood cell walls with much greater definition than is obtainable using conventional LM, with the added advantage of obtaining specific chemical information from considerable depths from the surface of tissues being examined through reconstruction of optically sectioned tissues [95,96]. CLSM has been most widely used to extract information on the intensity and distribution of the lignin component of cell walls using lignin autofluorescence or a reagent (e.g., acriflavin) staining enhanced lignin fluorescence [97]. Combining treatment with fluorescent antibodies can also provide specific information on the intensity and distribution of other cell wall components [97]. The relevance of the use of CLSM in the analysis of waterlogged woods has also been demonstrated [9]. Similarly, high-resolution images displaying lignin distribution can be acquired using UV microscopy, and when combined with spectrophotometry, topochemical information on lignin concentration and distribution can be obtained with high specificity [98]. The usefulness of UV microscopy has also been demonstrated in the analysis of waterlogged woods, particularly confirming that the residual material accumulating in wood tissues attacked by erosion bacteria is rich in lignin [99,100]. Access and close familiarity with the operation of the microscope and the accurate interpretation of results are major limitations for CLSM and UV microscopy. Embedding samples in a polymer and obtaining sections of a precise thickness is another limitation in UV microscopy work.

Confocal Raman microscopy (CRM) has been widely used for the chemical imaging of plant and wood cell walls at high resolution and specificity without the need for the tedious sample preparations associated with many other types of microscopy. This means that the samples can be analysed in their native state, and the concentration and distribution of cell wall components can be scanned across the cell wall [101,102]. Chemical imaging with CRM can yield valuable information on molecular changes occurring from biological, physical, and chemical treatments of wood. However, the application of CRM in the analysis of waterlogged archaeological woods is limited [103], mainly because of the difficulty in obtaining suitable sections from the highly degraded outer tissues of such woods. However, the analysis of partially degraded tissues from the region interfacing intact interior tissues holds promise. Another important limitation is accessibility to the instrument and expertise in its operation and the accurate interpretation of results, as only a few laboratories around the world house the facility and its takes years of experience to acquire the skills and knowledge. Not surprisingly, CRM work often involves collaboration. Vibrational microspectroscopy based on confocal Raman spectroscopy and FT-IR microspectroscopy [104] are powerful methods for in situ analysis of chemical composition in the context of cell wall microstructure, resulting from the chemical, mechanical, and biological modification of wood, and its application in the analysis of waterlogged woods should be explored.

As a high-resolution imaging tool, electron microscopes (SEM and TEM) have been widely used in wood and cell wall structure studies. In combination, these instruments provide most valuable information—SEM for imaging wood and wood structural components in 3D, and TEM for obtaining detailed textural information on cell walls and other cell structures, such as pits and pit membranes. Electron microscopy has also served well in wood biodegradation studies. As relevant to waterlogged archaeological woods, the confirmation of the presence of the bacterial degradation of cell walls benefitted from the initial electron microscopic studies, where the high-resolution imaging of wood undergoing degradation in a range of natural environments revealed the presence of unique and distinctive degradation patterns resulting from bacterial attack on wood cell walls. One of the patterns (bacterial erosion) can now be diagnosed using LM alone; however, the confirmation of the presence of bacterial tunnelling of the cell wall, particularly in

heavily degraded waterlogged wood tissues, requires the use of electron microscopy, particularly TEM [105]. TEM tomography has been used to obtain 3D information on wood cell wall organization [106] and has the potential to provide useful information on the architecture of waterlogged wood cell walls and the intricacies of microbial decay patterns, particularly in understanding the 3D nature of bacterial tunnels and the associated slime. TEM-immunocytochemistry, in combination with specific antibodies, is a powerful technique for understanding the presence and distribution of cell wall components and has also been applied to wood research [107]. The application of this technique in the analysis of archaeological waterlogged woods should be explored to understand chemical changes more precisely in cell walls during microbial degradation.

Cryo-TEM and cryo-SEM, involving the rapid freezing of plant tissues, enable cell walls to be examined in near-native state, although the application of these methods in wood biodegradation research has been limited [21]. The cryotechnique does hold promise for examining waterlogged archaeological tissues in various states of degradation for a more accurate analysis of the ultrastructural changes taking place. Buried and waterlogged woods are often infiltrated with inorganic substances, judging by the analyses of ash contents. Because such substances may adversely affect the conservation of waterlogged archaeological woods, it is advisable to assess their presence and distribution across the cell wall at a high resolution using SEM-EDS and/or TEM-EDS, the techniques widely used for elemental mapping in plant research, including wood science. However, although facilities are available in most advanced microscopy centres, these SEM- and TEM-based techniques are cumbersome and time-consuming. Therefore, the value of using these techniques for waterlogged woods has to be weighed against the benefit gained, as the ultimate goal of the analysis is developing most effective methods for conserving ancient waterlogged wooden objects.

Atomic force microscopy (AFM), which is a non-destructive, high-resolution scanning probe microscopy technique, has been widely used in plant and wood research, and is particularly suited to probing nano-scale cell wall architecture and the arrangement of cellulose microfibrils. Through creating nano-indentations into the surface of a material, AFM can be used to obtain information on cell wall properties, such as stiffness and hardness, as reviewed in Ref. [95]. As the technique can provide precise information on a specific region of the cell wall, it has been used to assess property improvements following wood treatment/modification. However, as AFM is a highly specialized tool, only a few microscopy centres around the world house this instrument. Therefore, its use in the analysis of waterlogged archaeological woods has to be weighed against the usefulness of information, particularly in the context of conservation. The quality of sample surface is another limitation, as the surfaces to be examined have to be absolutely smooth. However, for waterlogged woods, block face or ultrathin sections from polymer-embedded tissues can overcome this problem, but the possible shrinkage of the cell wall resulting from embedding has to be kept in mind.

Synchrotron X-ray microtomography is being increasingly used to study the 3D microstructure of wood. Because of the high sensitivity of this technique, small structural changes caused by physical, mechanical, and biological factors can be non-destructively quantified [108]. However, as access to synchrotron facilities is limited and the operation of the instrument and interpretation of the results require an in-depth knowledge and skills, the application of this method for evaluating waterlogged archaeological woods has to be carefully weighed against the benefits gained.

When using microscopy techniques to evaluate the structural and chemical state of archaeological waterlogged woods destined for conservation, it is important to be mindful that microscopy on its own does not provide a complete picture, and has to be combined with other methods, particularly for chemical assessment. Furthermore, microscopy can cover only a small sample size, and many measurements have to be performed to gain a better understanding of the state of bulk wood. Conventional analytical techniques, when combined with chemical-based spectroscopic methods, such as FTIR, enable a larger volume

of samples to be analysed, as well as precise chemical information to be obtained [109], aiding in the improved understanding of the chemical changes occurring during the microbial decomposition of waterlogged woods.

## 4. Decay Types

### 4.1. Fungal Decay

The fast-degrading basidiomycete fungi (white-rot and brown-rot fungi) are excluded from waterlogging environments mainly because of restrictive oxygen availability. Wood under such conditions is degraded by soft-rot fungi and bacteria. Under extreme oxygen-limiting conditions (anoxic conditions), even soft-rot fungi and tunnelling bacteria are excluded, and wood is exclusively degraded by erosion bacteria, suggesting that there is a critical threshold for oxygen concentration needed for soft-rot fungi and tunnelling bacteria, and that erosion bacteria are highly tolerant to reduced oxygen levels. It is generally considered that wood degradation by erosion bacteria under anoxic conditions is extremely slow; it is therefore not surprising that buried and waterlogged wooden objects have been found to survive centuries of exposure to such environments, as reviewed in refs. [4,6].

The mechanism of the soft-rot degradation of wood involves cavity formation within the cell wall (Type I) or cell wall erosion (Type II) [14]. Thus far, the majority of studies have reported the presence of Type I attack in waterlogged woods, except for a recent report [31] that mentions the occasional presence of Type II attack in addition to widespread Type I attack in a waterlogged wood. Therefore, the decay micromorphology will mainly focus on Type I attack, as relevant to diagnosing its presence in waterlogged woods.

#### 4.1.1. Soft Rot Type I

After penetrating into the S2 layer, the thickest part of the wood cell wall, hyphae produce chains of cavities with pointed ends (commonly diamond-shaped) during cell wall degradation. When fully developed, the cavities can readily be viewed using LM, particularly under the polarization mode. Thus, LM has proved to be a favourite tool for the analysis and diagnosis of soft rot attack on wood tissues [110] (Figure 3), because the resolution of the light microscope is sufficient for detection, and the samples can be rapidly scanned for the presence of soft rot. The cavities appear to be circular or near-circular in their form when viewed in transversely cut sections of the cell wall (Figures 3 and 4), and diamond-shaped in longitudinal sections. While both section types can be informative for diagnosing the presence of soft rot in waterlogged woods, longitudinal sections are more commonly used as they are informative in other important ways. For example, the orientation of cavity chains mirrors the orientation of cell wall microfibrils. The strict parallelism has been taken advantage of in determining the orientation of cellulose microfibrils based on the orientation of cell wall cavities [111–113]. The reason for the close relationship is not exactly clear, although it has been proposed that the alignment with microfibrils may trigger enzyme production. The fact that for the formation of soft rot cavities, lignin is not needed and cavities of the usual form can develop in cellulosic substrates suggests that it is the cellulose that influences cavity formation [114]. However, lignin concentration and type do affect the speed and extent of the degradation of wood cell walls by Type I soft rot, as reviewed in refs. [6,14]. For example, the highly lignified middle lamella persists even after the extensive degradation of the secondary cell wall, and other lignin-rich wood structures, such as ray tracheids and initial pit borders in *Pinus radiata* [115,116], and the S3 layer in conifers [14], are also highly resistant. Cavity formation is delayed in guaiacyl lignin-rich vessel cell walls [14]. The middle lamella and the S3 layer were found to be intact also in waterlogged archaeological wood cell walls attacked by soft rot Type I [117]. Thus, the presence of these soft-rot-resistant structures in wood tissues has implications for the survival of wooden objects exposed to waterlogging environments, as they can slow down the speed of cell wall degradation.

Whereas LM has proved adequate in the diagnostic assessment of the presence of soft rot Type I attack, the application of high-resolution tools, such as SEM and TEM, has

greatly increased our understanding of certain aspects of cavity formation and cell wall degradation not resolvable by means of LM. Confirmation of whether cavities can develop in the extremely thin S1 layer of normal wood cell walls and even thinner S3 layer is difficult to obtain using LM. TEM observations suggest that the S3 layer is too narrow for soft-rot fungi to produce cavities. The other factor may be the high lignification of this layer, particularly in conifers. However, using LM and SEM, soft rot cavities have been visualized in the S1 layer of compression wood [118], which is much wider than in the normal wood. It thus appears that soft rot cavity formation is influenced by both physical constraint (cell wall thickness) and the extent of cell wall lignification. TEM has greatly aided in revealing the fine structure of cavity-forming hyphae and in understanding the processes of cavity formation [119]. TEM, in combination with the specific staining of ultrathin sections, has also been useful in revealing the granular texture of the material that is present in the expanding cavities around the hyphae and which has been proposed to represent a mixture of slime, melanin, and modified lignin [14]. TEM has demonstrated that, in some cases, wood degradation is not confined to the usual concentric zone around the hyphae but extends irregularly to considerable distances from where the hyphae are present in the cell wall, an unusual pattern referred to as diffuse cavity formation [120].

SEM enables soft-rot-degraded wood tissues to be examined at much greater resolution than achievable using LM, and the samples can be scanned at a much faster rate than achievable using TEM. When degraded wood samples can be prepared in a way that prevents or minimizes the distortion/collapse of tissues (particularly common when extensively degraded), the sharp, well-defined images acquired enable large sample volumes to be examined to assess the state of degraded tissues. Obtaining sections using cryo-microtome for SEM observation is a suitable method for preventing/minimizing the collapse of even extensively degraded tissues, as illustrated by Blanchette et al. [121] in their SEM examination of soft rot Type I-attacked wood tissues from Antarctic huts built by the early explorers. Examining such tissues at a low kV (2–5) using a field emission SEM can further enhance the image quality. Furthermore, it is possible to undertake correlative LM, SEM, and TEM microscopy of consecutive sections obtained from the same volume of degraded wood tissues for complementary information, as achieved by Singh et al. [122], who used the same sections to examine the wood-adhesive interface using LM, CLSM, and SEM. The technical approach used by Blanchette et al. [121] for SEM imaging would be applicable to assessing the state of tissue disintegration in archaeological waterlogged woods, as the morphology of SR cavities can be well preserved even in extensively degraded tissues, where neighbouring cavities may have coalesced, resulting in large irregular voids. Even in such a state of degradation, the presence of the S3 layer and the middle lamella is indicative of the SR Type I attack, as observable in the SEM images in the work of Blanchette et al. [121] on extensively degraded tissues containing irregular voids in the cell wall. The soft rot Type I micromorphology described above should prove to be of value in diagnosing the presence of this type of attack in waterlogged archaeological woods even in advanced stages of degradation.

### 4.1.2. Soft Rot Type II

The majority of studies have found this type of fungal attack to be absent from waterlogged archaeological woods, except for a recent report that mentions its occasional presence in a waterlogged wood [31]. Type II soft rot attack is more common in hardwoods than softwoods. The cell wall is degraded along the exposed faces by way of erosion, with the degradation progressing towards the middle lamella. However, the middle lamella remains even when the secondary cell wall has essentially disappeared, suggesting the resistance of highly lignified cell wall regions, such as the middle lamella [30,123,124]. Also, the lignin-rich S3 layer in softwoods resists degradation in the initial stages of fungal attack. Both lignin concentration and type appear to affect cell wall degradation by soft rot Type II. This view is supported by the fact that hardwoods are more rapidly degraded than softwoods, likely because of the higher lignin content of the latter. Furthermore, hardwoods

with a high lignin content exhibit considerable resistance. The available information suggests that, in this type of soft rot attack, the cell wall is degraded by the enzymes released from the hyphae growing in the cell lumen [14]. The question of whether a close fungal contact with the cell wall is a pre-requisite for enzyme release has yet to be resolved. The fungal preference is for the S2 layer, the richest source of utilizable nutrients in the cell wall. The puzzling question, however, is how the S3 layer, which in softwoods is generally more highly lignified, is by-passed, knowing that this cell wall layer offers considerable resistance at least in softwoods, and is too compact for the large-size wood-degrading enzymes to diffuse through. The reader is directed to Anagnost [110] for other useful diagnostic features imaged using LM.

LM enables the rapid assessment of the presence of soft rot Type II attack in wood samples, including those of waterlogged ancient archaeological woods, and electron microscopy can offer the advantage of resolving fine details of eroding cell walls. Although at a cursory glance, the microscopic appearance of the pattern of soft rot II attack (erosion troughs/erosion channels) is similar to that of simultaneous white-rot degraders [30,110,115,125]. The presence of the middle lamella even in advanced stages of cell wall degradation [126] is a distinctive diagnostic feature of soft rot Type II attack, and this feature, in combination with the presence of erosion troughs, can be helpful in assessing the presence of soft rot Type II attack in ancient waterlogged woods. In simultaneous-type white rot attack, even the middle lamella is degraded and can disappear in part or entirely in later stages [30].

### 4.2. Microscopic Journey Leading to the Unravelling of Distinctive Bacterial Decay Patterns

While soft rot decay patterns are recognizable under LM, understanding that wood can also be degraded by bacteria had to await the application of electron microscopy, particularly TEM, for obtaining high-resolution images initially from decaying wood and wood products exposed to natural terrestrial and aquatic environments, as reviewed in Refs. [11,14,19], although the presence of bacteria in decaying wood has been recognized for a very long time, as reviewed in ref. [23], with the first LM image captured in 1950; see Ref. [23]. As we now know, the LM view represents a decay pattern resembling that produced during the bacterial erosion of lignocellulosic cell walls. However, it was not possible at that time to unequivocally relate the origin of the decay pattern to bacterial activity. This is because of the inadequate resolution of LM, and the lack of definition resulting also from inappropriate sample preparation. This, plus the fact that, in many early studies undertaken, the pure isolates of bacteria used were unable to degrade wood cell walls in their native state, as reviewed in Ref. [23], led researchers to believe that bacteria could not degrade unmodified wood. This also had implications for understanding the cause of the deterioration of wood subjected to prolonged waterlogging. It was believed that wooden objects subjected to prolonged burial and waterlogging deteriorated not from microbial factors but from chemical hydrolysis taking place over a long time, as reviewed in Refs. [28,127]. While chemical degradation cannot be entirely discounted, we now know that, in such environments, wood suffers from biological degradation, most prevalent being the degradation by erosion bacteria [6,9,15–17,19,26–28,127–130], the wood decay microorganisms most tolerant to oxygen limitation that is characteristic of waterlogged environments.

The application of TEM, in studies undertaken in the late 1970s and throughout the 1980s, created a flurry of activities reporting the bacterial degradation of wood samples obtained from a wide range of natural environments, as reviewed in refs. [14,19]. The confirmation that certain types of bacteria can degrade lignified wood cell walls (in their native state) in nature was achieved by combining the high-resolution capability of TEM with specific preparation techniques, the most important being the ultrathin sectioning of decaying wood samples after appropriate chemical fixation and subsequent embedding in a polymer resin, taking advantage of the information already available from numerous electron microscopic studies of plant and wood materials. The high-resolution and

high-definition images of cell wall regions undergoing degradation captured using TEM provided evidence of a close spatial relationship of bacteria with such cell wall regions. The degradation patterns resulting from bacterial activity were distinctive and very different from the well-known fungal degradation patterns. The patterns were placed into two distinct categories: bacterial degradation of the wood cell wall via tunnelling action (with the pattern named 'bacterial tunnelling' and the bacteria causing this type of decay 'tunnelling bacteria') and bacterial degradation of the wood cell wall via cell wall erosion (with the pattern named 'bacterial erosion' and the bacteria causing this type of decay 'erosion bacteria') [131–135].

Although there are variant terms to describe the above patterns, such as 'burrowing' instead of tunnelling, 'tunnelling' and 'erosion' have been widely accepted terms in use refs. [4–6,14,19,39,136,137]. The use of SEM yielded complementary information and strengthened the conclusion drawn from TEM observations, particularly for bacterial erosion-type attacks, where spectacular images of erosion troughs containing bacteria could be acquired using SEM. Furthermore, the techniques developed to faithfully reproduce the unique bacterial degradation patterns in the laboratory by exposing wood pieces to mixed bacterial cultures firmly established the proposed classification of bacterial decay into two distinctly different forms. Correlative microscopy involving the TEM and SEM examination of the sections (or the sections from the same tissue volume) that had been examined using LM aided in the identification of the bacterial decay patterns using LM alone, particularly the bacterial erosion pattern, the most prevalent form of attack in waterlogged woods, such as the foundation piles supporting historic buildings and ancient shipwrecks buried deep in ocean sediments [6,28]. The ability to recognize the bacterial erosion pattern using LM alone [138] revolutionized the work on ancient waterlogged woods, as the diagnostic microscopic examination could be carried out without reliance on the tedious and demanding technical processes of electron microscopy. The micromorphology of bacterial decay patterns will now be briefly described, with a focus on the relevance to diagnosing the decay types in waterlogged woods.

To summarize, although the presence of soft rot cavities can be readily resolved using LM, soft rot cavities and the process of cavity formation can be best studied using combined LM, SEM, and TEM techniques (Figure 5). TEM is the most suitable high-resolution tool for diagnosing bacterial tunnelling-type attacks in waterlogged woods, examining tunnelling bacteria and tunnel micromorphology, and understanding the tunnelling process (Figure 5). While the combined use of SEM and TEM can provide detailed information on erosion bacteria, erosion troughs, the process of bacterial erosion, and residual material, LM alone is now being routinely used to diagnose the presence of bacterial erosion-type attacks in waterlogged woods (Figure 5). TEM, LM, UV microscopy, fluorescence laser confocal microscopy, and confocal Raman microscopy have all been used to characterize the chemical composition of the residual material, with the latter three providing more specific information.

### 4.2.1. Bacterial Tunnelling

After gaining entry into wood initially via rays, tunnelling bacteria invade lignified tissues where the bulk of the nutrients is stored in their cell walls. They colonize the cell lumen by degrading pit membranes common to neighbouring cells, which consist of cellulose, pectin, and hemicellulose [139,140], and can be readily degraded by these bacteria. Tunnelling bacteria then penetrate into lignified cell walls, gaining entry into the S2 layer by making a hole into the S3 layer sufficiently large for penetration. In this function, the ability of tunnelling bacteria to change their shape assists them in overcoming the physical and chemical constraints of the S3 barrier. This layer is highly lignified, particularly in conifers, such as radiata pine [80]. The processes of bacterial attachment to the exposed face of the S3 layer, with the help of the extracellular slime that tunnelling bacteria abundantly produce, and the subsequent bacterial entry into the cell wall, facilitated by shape changes, have been well documented and reviewed [6,14,19].

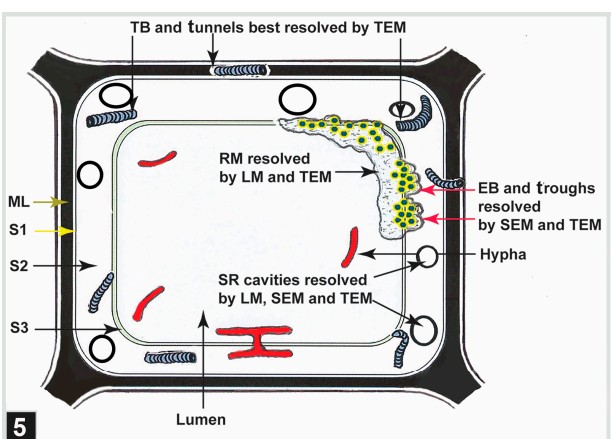

**Figure 5.** A diagram illustrating the types of microbial attacks reported for ancient waterlogged woods. Microscopic techniques used for identification and understanding of the decay patterns are indicated. EB, erosion bacteria; SR, soft rot; TB, tunnelling bacteria; ML, middle lamella; S1, S2, S3, secondary cell wall layers.

Once within the S2 layer, tunnelling bacteria develop microcolonies and degrade the cell wall through tunnelling action (Figure 5). As the bacteria tunnel through the cell wall, they repeatedly divide, with daughter cells moving and tunnelling in different directions (Figure 5), without a regard for microfibril orientation, unlike in the soft rot Type I attack, where fungal cavities faithfully follow the direction of microfibrils. The early stages of attack gives the impression of a pattern with tunnel branches radiating from a central point (Figure 6), reminiscent of tree branching. Although it is possible to detect the presence of bacterial tunnelling as fine radiating striations using LM in the early stages of attack, TEM is the ideal tool to clearly resolve the unique, intricate structure of tunnels (Figure 6), confirming the presence of bacterial tunnelling-type attacks, particularly in the advanced stages of cell wall degradation and where mixed microbial attacks (Figures 5 and 7) involving soft rot, tunnelling bacteria, and erosion bacteria in differing combinations are present, a feature common in wet terrestrial and aquatic natural environments [15,27,38,39,44,105,117,137,139–143].

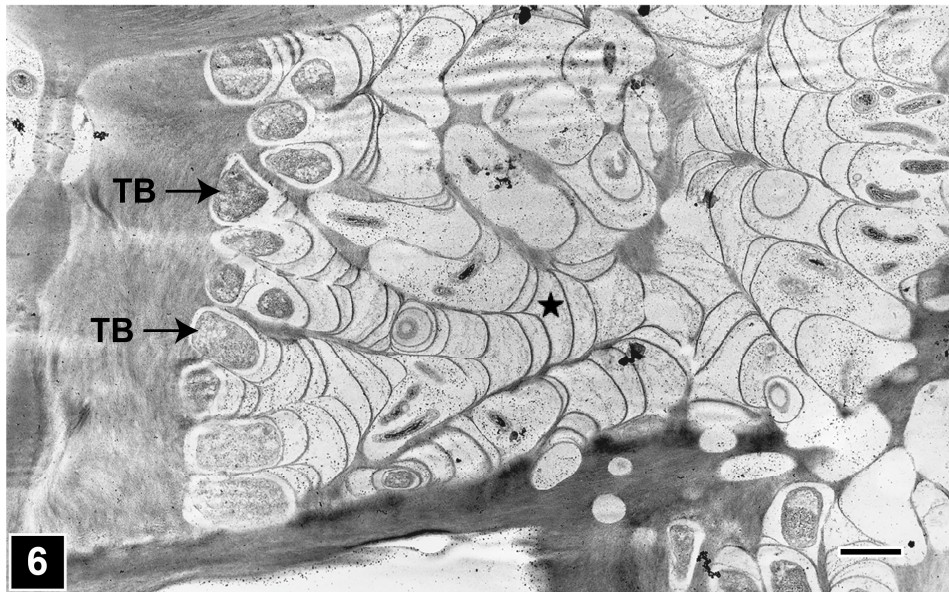

**Figure 6.** TEM micrograph showing the repeated branching of tunnels (asterisk) radiating from a central point in a wood cell wall. TB, tunnelling bacteria. Scale bar = 2 μm. The image is reproduced from Singh et al. [22].

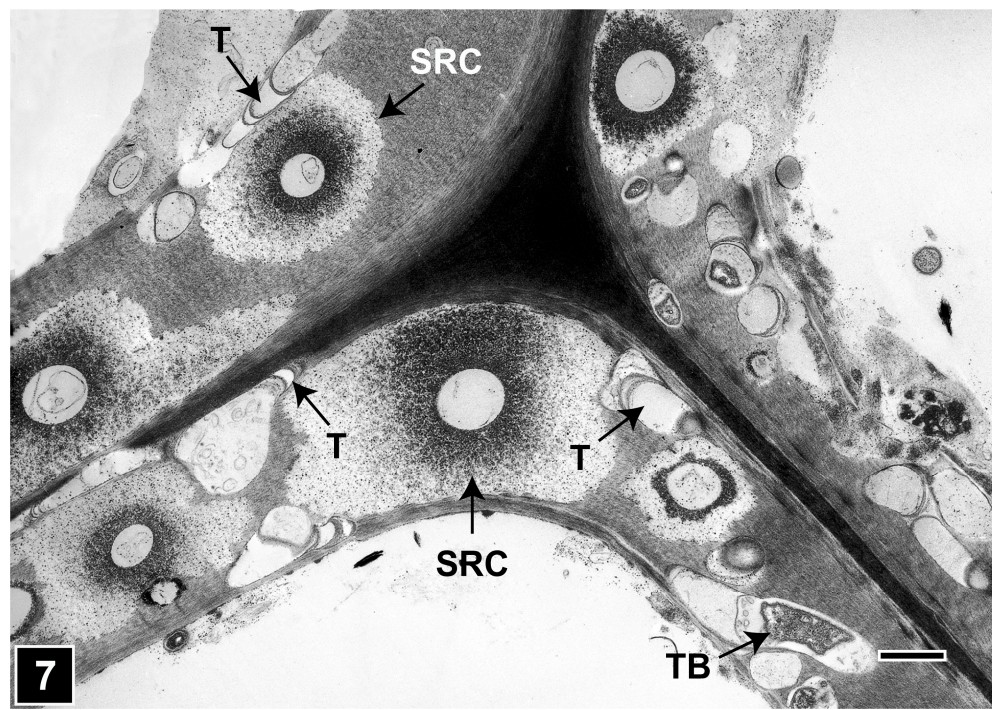

**Figure 7.** TEM micrograph showing the presence of mixed microbial attack on wood cell walls, involving soft rot Type I (diffuse degradation pattern) and bacterial tunnelling in the same tracheid cell wall. SRC, soft rot cavity; T, tunnels; TB, tunnelling bacteria. Scale bar = 2 µm. The image is reproduced from Singh et al. [44].

From the perspective of detecting the presence of bacterial tunnelling-type attacks in ancient waterlogged woods, the unique architecture of the tunnel, with periodic slime bands (cross walls), is the most important diagnostic feature (Figures 6 and 8). The aspects of slime function in bacterial movement (gliding) within the cell wall, band formation within the tunnels, the relationship of the form of the bands (crescent shape), as visible in transverse sections of wood cell walls, to the direction of tunnelling, bacterial form, and ultrastructure, and the micromorphological pattern of cell wall degradation have been described in detail in many earlier studies, and the reader is referred to pertinent reviews [6,11,14,19,50,105]. TEM has played a crucial role in understanding the intricate structure of the tunnel and the process of cell wall degradation. Admittedly, this high-resolution tool is best suited to confirming the presence of bacterial tunnelling-type attacks in the advanced stages of cell wall degradation, when the integrity of the tunnel structure may be compromised and only scant remnants of tunnel bands may be present [27,38,105]. TEM has thus played a vital role in diagnosing the presence of bacterial tunnelling-type attacks in waterlogged ancient woods, particularly where outer wood tissues are extensively degraded [105]. SEM has also served well to confirm the presence of bacterial tunnelling in ancient waterlogged woods, but only when tunnel bands are present and clearly visible [32]. The other helpful diagnostic feature is the degradation of the middle lamella (Figure 9), the most highly lignified part of the cell wall which is resistant to soft-rot and erosion bacteria, judging by the TEM images revealing the frequent penetration of inter-corner middle lamella regions and the occasional penetration of the cell corner middle lamella, the most highly lignified region of normal wood cell walls as determined using a range of techniques, including UV microscopy, confocal fluorescence microscopy, and the TEM of ultrathin sections after staining with $KMnO_4$, as reviewed in Refs. [49,56,57].

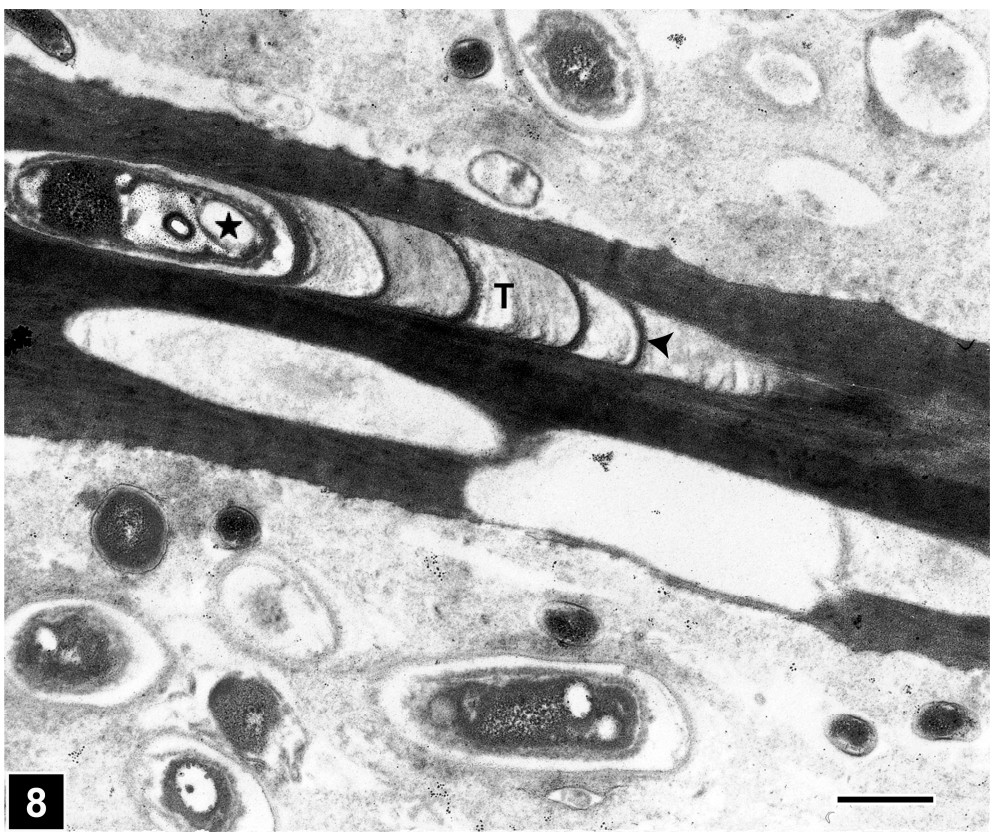

**Figure 8.** TEM micrograph showing the presence of a tunnel (T) with characteristic crescent-shaped periodic slime bands (arrowhead) in a tracheid cell wall. The asterisk marks a tunnelling bacterium within the tunnel. Scale bar = 2 µm. The image is reproduced from Singh et al. [44].

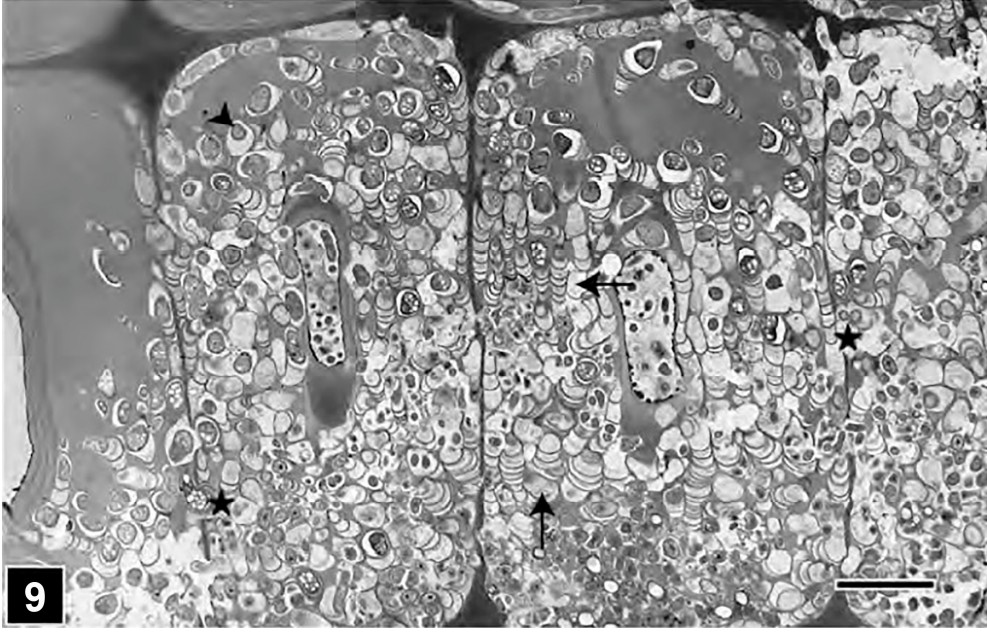

**Figure 9.** TEM micrograph showing the extensive degradation of *Homalium foetidum* (a high-lignin hardwood) fibre cell walls by tunnelling bacteria (arrowhead). All cell wall regions, including the highly lignified middle lamella (asterisks), are tunnelled. The direction of tunnelling (arrows) is variable. Scale bar = 8 µm. The image is reproduced from Singh et al. [144].

### 4.2.2. Bacterial Erosion

From the perspective of understanding the deterioration and conservation of water-logged archaeological woods and wooden objects, diagnosing the presence of bacterial erosion-type decay is undoubtedly most crucial, as this type of attack is prevalent and often the only form of microbial decay present in woods exposed for prolonged periods to anoxic burial and waterlogging environments [4,6,28]. Early LM studies of buried and waterlogged wooden objects, such as foundation piles supporting historic buildings, as reviewed in refs. [7,145], while showing the images of a decay pattern which we can now recognize as caused by bacterial erosion [7,145], were not conclusive due to the limitation of LM in clearly resolving a spatial relationship of the bacteria to the cell wall regions under degradation. The most fitting example is the LM micrograph taken in 1950 of a foundation pile sample, reproduced in the review by Schmidt and Liese [23], which shows a decay pattern resembling bacterial erosion, as we now know. The confirmation that bacteria can decay wood via cell wall erosion (bacterial erosion) had to await the application of SEM and TEM, which, in combination, clearly resolved the spatial relationship of erosion bacteria with the cell wall regions undergoing degradation and the intricate micromorphological features associated with the degradation process [26,134,146] (Figures 10 and 11).

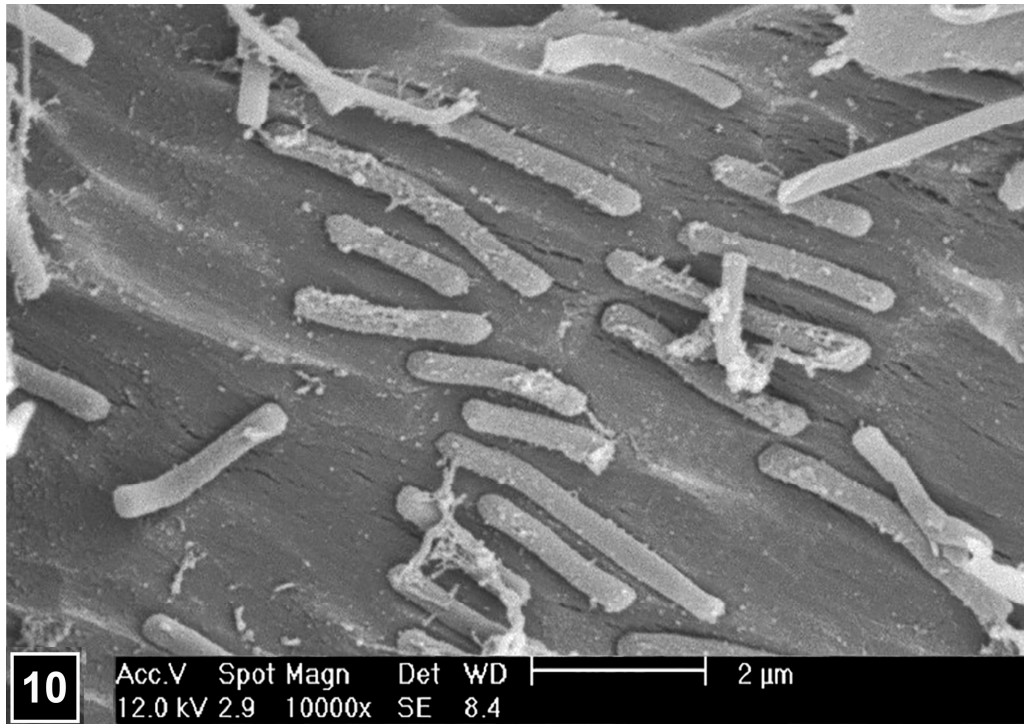

**Figure 10.** SEM micrograph of wood cell walls undergoing bacterial erosion, with rod-shaped erosion bacteria present in the erosion troughs. The micrograph is courtesy of Professor Charlotte Björdal, University of Gothenburg, Sweden.

The most important distinguishing feature of bacterial erosion is the presence of distinctive troughs into the eroding face of the cell wall and the close spatial relationship of bacteria with the cell wall under degradation, as viewed in both transverse and longitudinal sections of wood samples (Figures 10 and 11). In the SEM images of longitudinal sections, the cell wall erosion appears in the form of long, discrete, parallel troughs, each containing rod-shaped bacteria positioned linearly [134,146] (Figure 10). In TEM images of transverse sections, the erosion troughs appear as a series of crescent-shaped notches into the eroding face of the cell wall, each with a circular-appearing bacterium closely fitting into the notch [26,134] (Figure 12). The erosion bacteria and erosion troughs are invariably aligned with the long direction of cellulose microfibrils, as revealed using SEM (Figure 10) and also in favourable views captured using TEM [133], an aspect of the decay process common

also to soft rot Type I attack, where the long direction of soft rot cavities corresponds to the orientation of the microfibrils. How this relationship influences the bacterial erosion process Awould be worth exploring, particularly from the perspective of understanding the molecular mechanism underlying cell wall erosion. The observations made on samples from different wood types and exposure environments and conditions suggest that the appearance of the micromorphological features developed during the bacterial erosion of cell walls is remarkably consistent, although there are reports of the effect of oxygen concentration on the speed of cell wall degradation, with the slowest degradation taking place at anoxic or near-anaerobic conditions, typical of the burial of wood deep in ocean sediments from where many precious wooden objects have been uncovered.

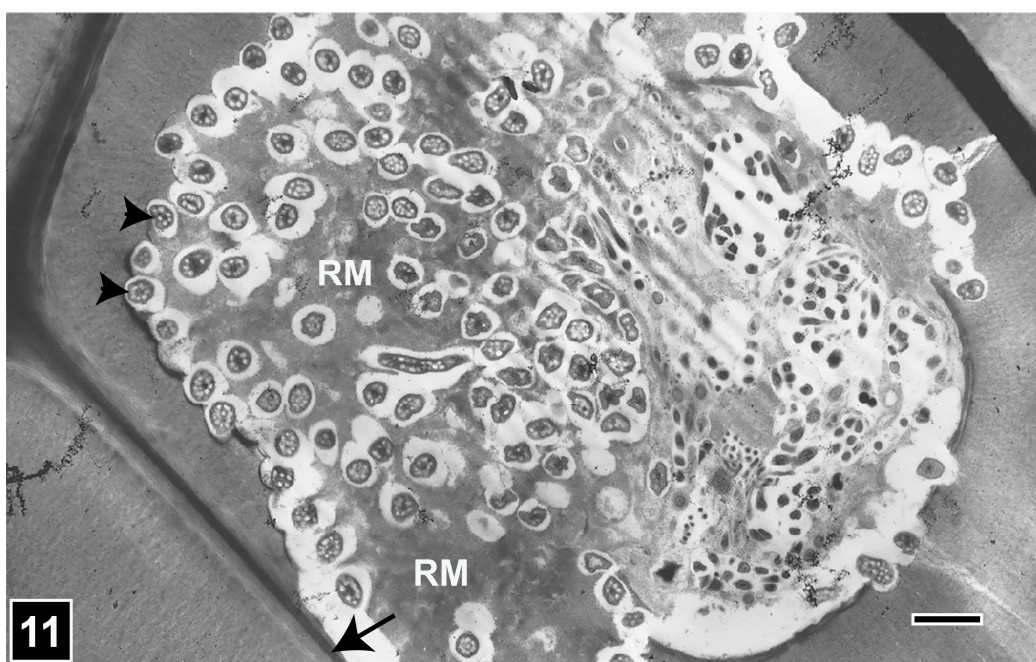

**Figure 11.** TEM of a transversely cut tracheid cell wall attacked by erosion bacteria (EB). EB are present opposite crescent-shaped erosion troughs formed in the exposed face of the secondary wall (arrowheads). The middle lamella is not degraded (arrow). RM, residual material. Scale bar = 4 μm. The image is reproduced from Singh et al. [22].

Other features that define bacterial erosion, such as the role of bacterial slime in the attachment and maintenance of bacterial position in contact with or close vicinity to the wood cell wall, the role of bacterial slime in bacterial gliding on the cell wall surface and within the trough as the erosion deepens, bacterial cell morphology and ultrastructure, the resistance of highly lignified cell wall regions and wood structures to erosion bacteria, and the presence of a residual material in degraded cell wall regions, have been described in detail in many publications and reviews of studies undertaken on wood samples exposed to various environments, including waterlogged woods [4,6,9,14,15,19,22,26,28,30,44,128,134,145], and only those relevant to diagnosing bacterial erosion in wooden objects subjected to prolonged waterlogging will be described here.

The most important relevant features are the presence of the middle lamella even in most advanced stages of decay (Figure 11), when all of the secondary cell wall may have been degraded; and the presence of residual material (Figures 11 and 12), which is a particularly prominent feature of extensively degraded wood tissues. It is worth noting that, in recent times, the use of LM, taking advantage of the information available from the initial TEM studies unambiguously defining features that are consistently characteristic of bacterial erosion, has greatly accelerated in diagnosing the presence of bacterial erosion in waterlogged woods (Figure 11). LM is now being routinely used as a standalone tool for this purpose [7,145], circumventing the tedious and time-consuming sample prepa-

ration for SEM and TEM examination, and making it possible to rapidly examine large volumes of samples necessary for properly evaluating the state of deterioration of uncovered waterlogged archaeological wooden objects for conservation. Initially, TEM proved useful in revealing the texture and chemical nature of the accumulated residual material (Figures 11 and 12), when ultrathin sections were examined after $KMnO_4$. In more recent times, other microscopy methods, such as the LM of sections stained with lignin-contrasting stains (Figure 13), confocal laser scanning microscopy (Figures 14 and 17), UV microscopy, and confocal Raman microscopy [9,99,100,145,147], have provided complementary information, confirming the conclusions drawn from TEM studies that the residual material is largely lignin and lignin products. These microscopies, particularly UV, confocal Raman, and confocal laser scanning microscopy [98,101,102,147–152], are powerful tools that can provide chemical information at a high spatial resolution, and thus have been used to scan the residual material as well as across the cell wall for the presence of lignin and variability in lignin concentration at a submicron level. Furthermore, the information obtained from chemical microscopy strongly supports the chemical characterization of wood degraded by erosion bacteria, including waterlogged archaeological woods, using other techniques, which suggest that, during bacterial erosion, carbohydrates are preferentially removed, with little or no effect on lignin [34,109,138]. Thus, the progress in evaluating the presence of bacterial erosion in waterlogged archaeological woods was made possible by the application of suitable electron microscopy tools and techniques to clearly resolve the intricate features of the decay pattern, which served as the diagnostic base reference for subsequently examining large volumes of wood samples rapidly and with simpler preparations using LM.

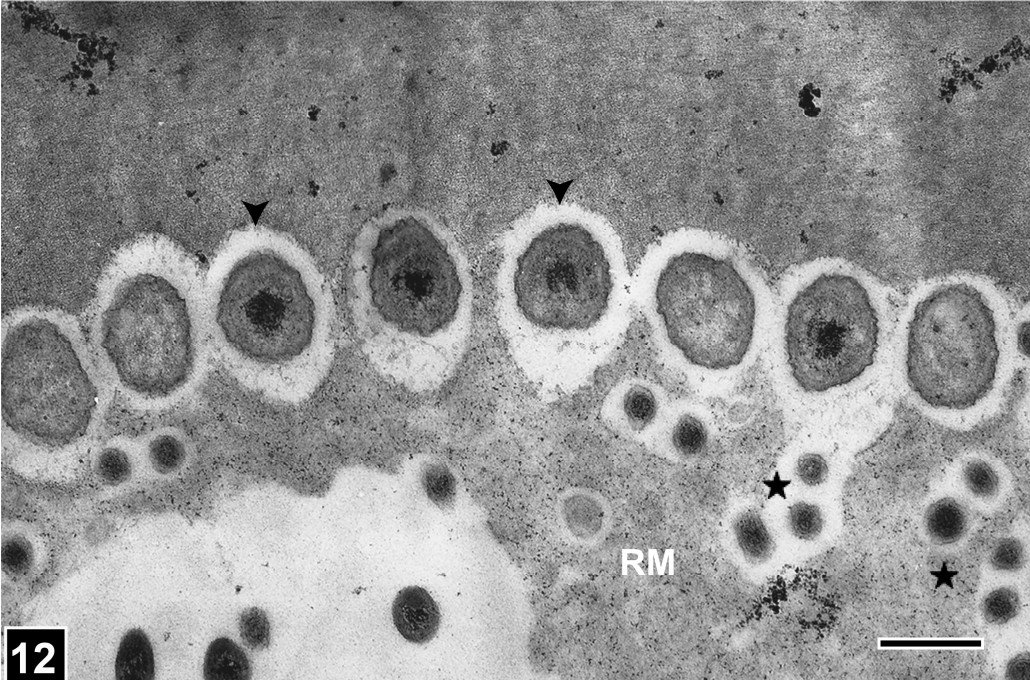

**Figure 12.** TEM of a transversely cut tracheid cell wall attacked by erosion bacteria (EB). EB are positioned opposite erosion troughs and display a near-circular form (arrowheads). The much smaller bacteria (asterisks) within the residual material (RM) are secondary degraders (scavenging bacteria). Scale bar = 2 μm. The image is reproduced from Singh et al. [44].

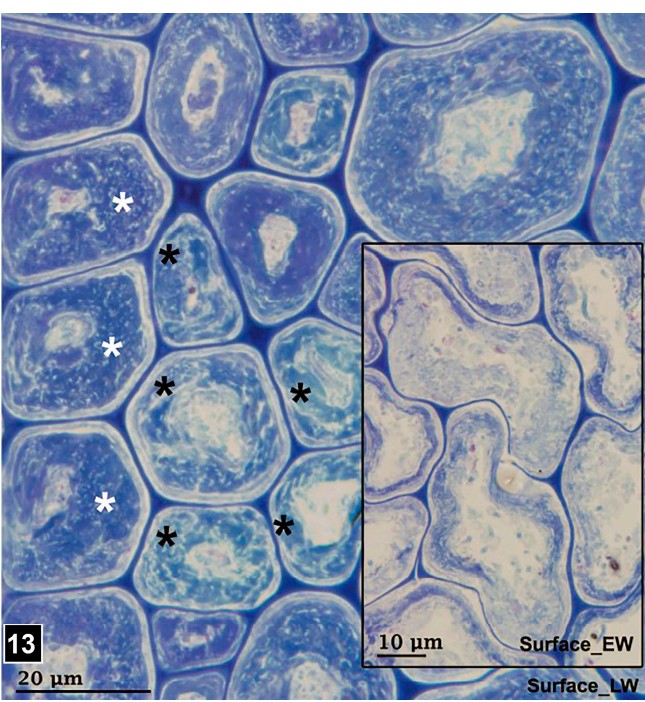

**Figure 13.** LM of transversely cut sections from the outer part of a waterlogged archaeological wood (Daebudo ship), stained with toluidine blue. The secondary cell wall of tracheids is extensively degraded, but the middle lamella is resistant. The earlywood (EW) and latewood (LW) cells are filled with a residual material, which appears more intensely stained in some cells than others (white vs. black asterisks), reflecting compositional heterogeneity. The images are reproduced from Cha et al. [9].

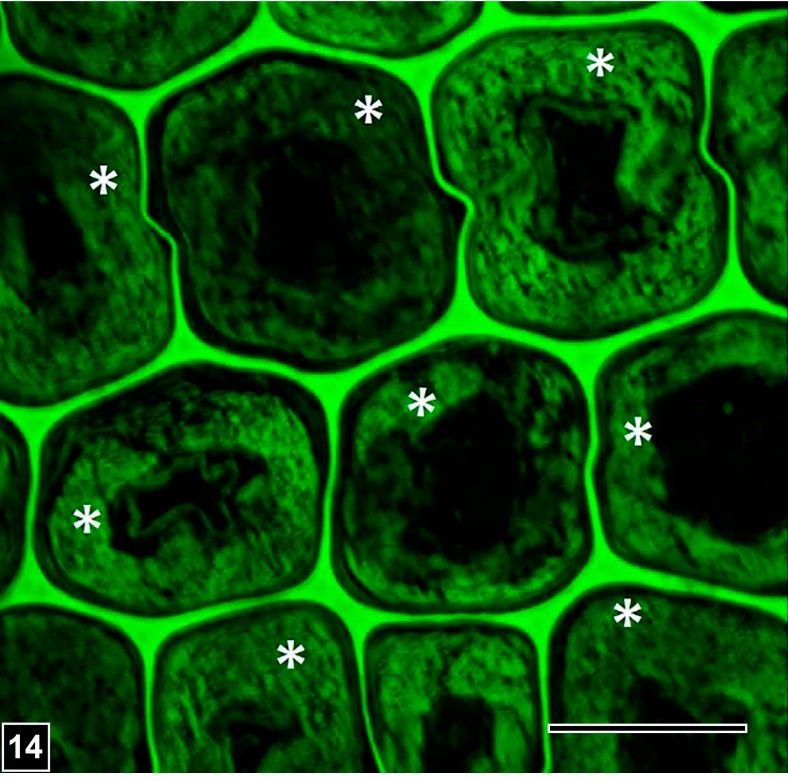

**Figure 14.** Confocal laser scanning micrograph (CLSM) of a transversely cut section from a waterlogged archaeological wood degraded by erosion bacteria. Based on acriflavine-enhanced lignin fluorescence, the strong fluorescence displayed by the residual material (asterisks) reflects its high lignin content. Scale bar = 15 μm. The image is reproduced from Singh et al. [6].

## 5. Highly Lignified Cell Wall Regions and Wood Structures Influence Microbial Degradation, as Revealed by Microscopy: Relevance to the Survival of Buried and Waterlogged Archaeological Wooden Objects

Because erosion bacteria are the predominant degraders of historically and culturally important archaeological woods exposed for prolonged periods to anoxic waterlogging environments, and the longevity of such woods depends on the speed of cell wall degradation, the information is reviewed only in relation to bacterial erosion. In regard to the resistance of highly lignified cell wall regions to erosion bacteria, more information is available on softwoods than hardwoods. The resistance of the following highly lignified cell wall regions and wood structures to erosion bacteria may make important contributions to the survival of waterlogged archaeological woods.

### 5.1. Middle Lamella and the S3 Layer

The middle lamella and the S3 layer of softwood tracheids are more highly lignified regions of the cell wall compared to the S2 layer, with the middle lamella being the most highly lignified region. For example, in *Pinus radiata* tracheids, lignin concentration in the corner middle lamella is estimated to be around 81% [153], and in the S3 layer around 53% [68]. In all microscopic studies undertaken of wood tissues attacked by erosion bacteria, as reviewed in refs. [14,19], including archaeological waterlogged woods [9,28,30], the middle lamella is always present (Figures 11 and 15), and the resistance of this cell wall region is attributable to its very high lignin content. Owing to the retention of the middle lamella in its entirety and the presence of much of the S3 layer (Figure 15), waterlogged wood tissues even in the advanced stages of bacterial erosion often retain their original form, thus facilitating the conservation process. However, such tissues are prone to collapse when subjected to burial load/pressure, which can render conservation problematic.

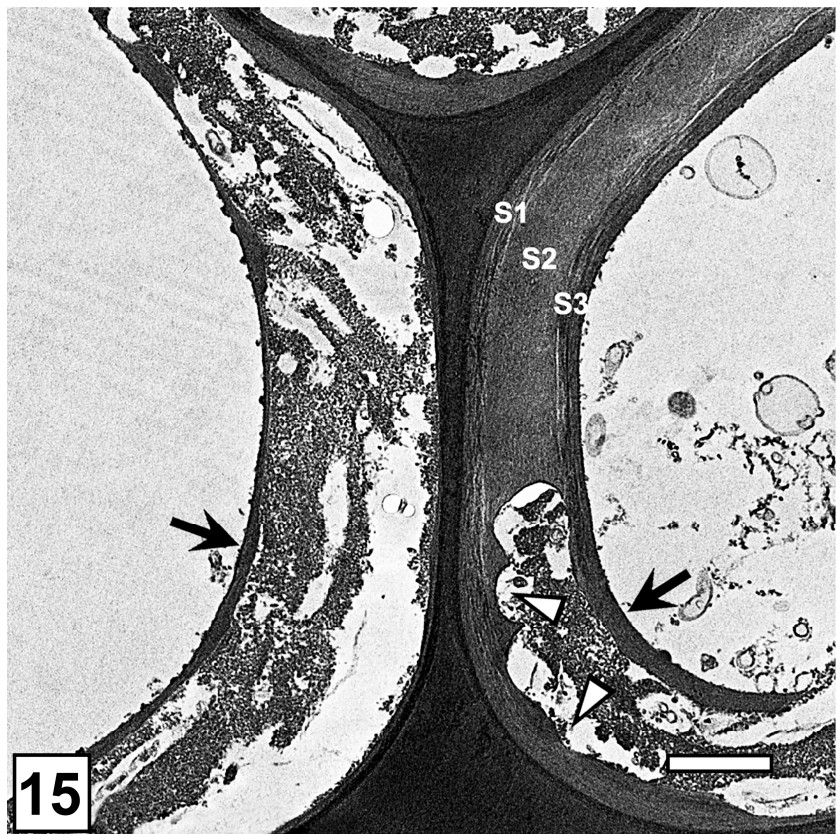

**Figure 15.** TEM of a transversely cut section from a waterlogged archaeological wood degraded by erosion bacteria. The micrograph shows the presence of the S3 layer despite the extensive degradation of the S2 layer of tracheid cell walls. Scale bar = 2 μm. The image is reproduced from Cha et al. [9].

### 5.2. Outer Part of the S2 Layer (S2$_L$) in Compression Wood

The outer part of the compression wood is more highly lignified than the inner S2 layer, with the magnitude depending upon the severity of compression wood [40,87]. Compression wood ranges from very mild forms to severe forms, the former often resembling a form typical of normal wood tissues, which require special microscopic preparations for identification, such as TEM, UV, and fluorescence microscopy, based on the assessment of lignin concentration, particularly where greater lignification of the S2 cell wall is confined to the cell corner region, as reviewed in ref. [40]. The S2$_L$ layer in the severe compression wood is highly resistant to bacterial erosion, as demonstrated for waterlogged archaeological woods [29], and even in mild forms, the most highly lignified cell corner regions of the S2 wall exhibit considerable resistance to this form of bacterial attack [38,39]. Perhaps the most fitting example is the discovery in Germany of the virtually intact Schöningen wooden spears, which had been ingeniously constructed and used by early hominids for hunting and had been buried in an anoxic environment for nearly 400,000 years. A TEM study explained the presence of mild compression wood as the main reason for the very long life of the spears in the burial environment, as the compression wood tissues resisted bacterial erosion, with degradation found only in the surface tissue layers [25] (Figure 16).

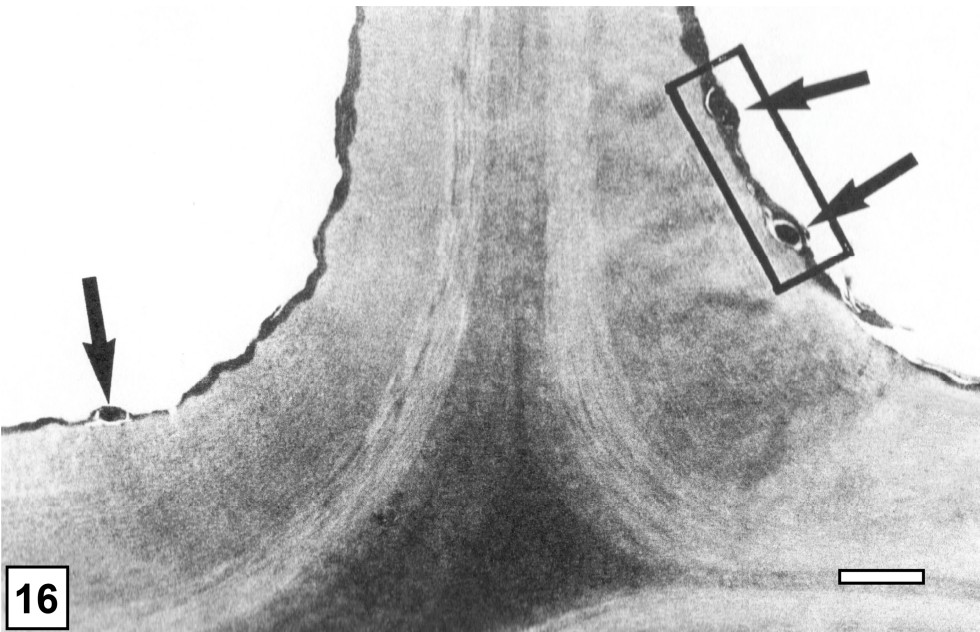

**Figure 16.** TEM of a transversely cut section through parts of mild compression wood tracheids from a Schöningen spear. The cell wall degradation by erosion bacteria (arrows) is confined to the innermost part of the cell wall (boxed region). Scale bar = 1 μm. The image is reproduced from Schmitt et al. [25].

### 5.3. Initial Pit Borders

Based on the examination of ultrathin sections stained with KMnO$_4$, TEM studies of wood exposed to a range of water-saturated and waterlogged environments [41,148], including ancient waterlogged woods [9,29,44,154], have provided evidence that initial pit borders in gymnosperm-bordered pits are resistant to bacterial erosion. The high lignification of the initial pit border is the likely reason for its resistance, judging by its distinctly greater electron density compared to other regions of the pit border in KMnO$_4$-stained sections and the high fluorescence of the pit border after the acriflavine staining of sections (Figure 17). Although the exact lignin content of the initial pit border has not been determined, this structure develops prior to secondary wall formation, and is considered to be similar to the primary wall [155]. Assuming the architecture of the initial pit border to be similar to the primary wall prior to lignification, the level of lignification of the initial

pit border is also likely to be similar to that of the primary wall, which is a highly lignified region of the cell wall [49] and resistant to bacterial erosion [22,26]. Thus, the presence of the primary pit border is relevant to the longevity of waterlogged archaeological woods exposed to anoxic environments, as it can prevent erosion bacteria from traversing the entire pit border [44], affecting the speed of cell wall degradation.

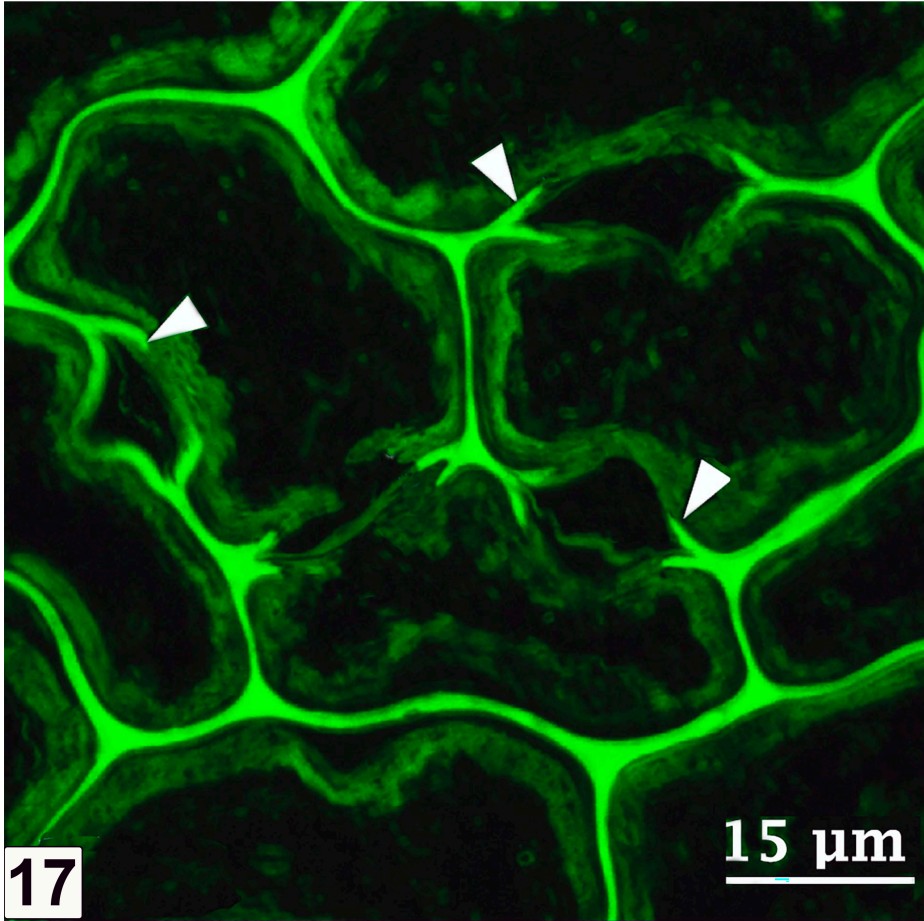

**Figure 17.** CLSM of a semi-thin section from a waterlogged archaeological wood degraded by erosion bacteria. The micrograph from the section, which was stained with acriflavine to enhance lignin fluorescence, shows the resistance of initial pit borders to bacterial erosion (arrowheads). The image is reproduced from Cha et al. [9].

*5.4. Warts and Vestures*

Warts are localized protuberances, extending from the inner face of the secondary cell wall. They are common to gymnosperm tracheids, as reviewed in ref. [156] (Figure 1), but can also be present in dicotyledonous woods [157]. Warts are highly variable in their size and form, and can be best viewed using SEM. TEM, which can readily resolve warts, has been widely used to examine them in earlier studies, e.g., [156]. Although appearing to be confluent with the underlying secondary wall, TEM examination has shown warts to differ from the secondary wall in their texture and composition. Compositionally, warts mainly contain lignin and some hemicellulose. Studies employing microscopic and chemical extraction techniques suggest that warts and the warty layer are more highly lignified than the underlying secondary wall [6,44,158–160], which is consistent with the conclusion drawn from TEM observations of $KMnO_4$-stained sections [6,9,44]. In TEM images captured from ultrathin sections stained with $KMnO_4$, warts exhibit greater electron density compared to the secondary cell wall from which they are elaborated (Figure 1). $KMnO_4$ has been widely used to contrast lignin in plant and wood cell walls [67,116,161–163] because of its specificity for lignin [164]. Further support comes

from studies employing chemical treatments to selectively remove lignin, such as the work of Kanbayashi and Miyafuji [165], where the treatment of *Cryptomeria japonica* wood with the ionic liquid 1-ethylpyridinium bromide led to the removal of warts. There are suggestions that lignin in warts is also more condensed, another factor that can enhance the resistance of warts to microbial degradation. The smoother texture of warts as visualized using TEM points to a lack of cellulose. Evidence for the presence of xylans in *C. japonica* normal and compression wood warts came from an immunolabelling study undertaken by Kim et al. [166].

Warts are often associated with an electron-dense thin layer applied to the inner face of the secondary cell wall, and together these structures are referred to as 'warts and the warty layer'. The presence of warts and the warty layer is very relevant to the longevity of ancient waterlogged woods exposed to anoxic environments. These structures can affect the speed of degradation as the resistant warty layer has to be breached by erosion bacteria first, before they can make contact with the nutrient-rich S2 layer. Several reviews and TEM studies of such woods have provided evidence of the presence of warts and the warty layer despite the extensive degradation of the underlying S2 layer by erosion bacteria [9,44,154], with the resistance being attributable to the high lignin content of warts and the presence of an extractive coating.

Vestures are protuberances from the inner part of the secondary cell wall, projecting into the cell lumen [167,168]. Bailey [169] was the first wood scientist to view and describe vestured pits using LM. However, the resolution of LM is inadequate to characterize the morphology of individual vestures and their distribution within pits with which they are associated. The use of TEM alone and in combination with SEM and other analytical methods has provided a wealth of information on the architecture, composition, and micro-morphological appearance of vestures [144,159,170–176]. SEM has been the high-resolution imaging tool of choice, as it is particularly suited to examining complex formations and arrangements of vestures in pit chambers [175], and the spectacular images obtained are not only aesthetically pleasing but are also of diagnostic value. TEM provides complementary information that improves our understanding of the relationship of vestures to the parent secondary cell wall, in addition to providing qualitative information on the lignin concentration of the vesture wall relative to the secondary wall. TEM can clearly resolve even the tiniest vestures present in wood tissues, in addition to demonstrating that vestures are confluent with the secondary cell wall [174], particularly in inter-vessel pits where vestures are most numerous, obscuring the visibility of a physical connection under SEM. Studies employing a range of chemical, histochemical, and electron microscopic techniques, including TEM observations of $KMnO_4$-stained ultrathin sections, suggest that vestures, in many cases, are highly lignified structures [144,159,172]. Watanabe et al. [173], in their compositional study of vesture walls in *Eucalyptus* species, using UV microscopy, TEM, FE-SEM (field emission SEM), alkali (NaOH) extraction treatment, and PATAg (periodic acid-thiocarbohydrazide silver proteinate) treatment, suggested that the vesture wall in *Eucalyptus* mainly consists of polyphenols and polysaccharides.

Judging by the often irregular form of vestures and their polyphenolic composition, it can be assumed that vestures, like warts, can play a role in extending the life of ancient woods subjected to prolonged exposure to anoxic burial and waterlogging environments. However, studies of the microbial deterioration of such woods are limited. In a TEM study of the wood from a Polynesian canoe (aged about 1000 years), recovered from a burial environment along the coast of a Pacific island, Donaldson and Singh [177] found that the wood had been exclusively attacked by erosion bacteria and reported the presence of vestures despite the extensive degradation of wood tissues. The preservation of vestures was attributable mainly to the presence of a covering resistant cell wall layer around the vesture wall matrix [177,178].

*5.5. Highly Lignified Ray Parenchyma*

Rays form the primary pathways for microbial entry and the colonization of wood tissues, primarily because parenchyma cells in rays are a rich source of readily utilizable nutrients for microorganisms, and ray parenchyma cell walls are normally not lignified. The degradation of ray parenchyma cell walls facilitates the entry of wood-degrading microorganisms into the tissues which form the bulk of wood (tracheids, vessels, and fibres) and where many of the nutrients are stored in their cell walls, forming a complex molecular network. However, in some wood species, such as *Pinus sylvestris*, parenchyma cells in some rays are highly lignified. In a microscopic study to investigate the cause and extent of the degradation of a bulwark which had been constructed from *Pinus sylvestris* wood in early 1100 in the lake Tingstäde Träsk on the island of Gotland, Sweden, LM operating under polarization mode provided evidence of the complete resistance of highly lignified ray parenchyma to erosion bacteria [154]. Whereas non-lignified ray parenchyma lacked birefringence, suggesting the loss of crystalline cellulose, lignified rays displayed a high birefringence. TEM confirmed LM observations, in addition to providing evidence of the extensive degradation of fibre cell walls [154]. These observations prompt us to suggest that the presence of highly lignified rays may contribute to the longevity of ancient waterlogged woods subjected to anoxic environments by forming a barrier to wood colonization by erosion bacteria.

**6. Heartwood and Heartwood Extractives Resist Microbial Degradation, as Revealed by Microscopy: Relevance to the Survival of Buried and Waterlogged Archaeological Wooden Objects**

Heartwood formation in trees is a natural, genetically programmed process evolved as a defence mechanism to deter microbial invasion into living trees. The understanding of heartwood formation and the amount and chemical nature of the extractives that impregnate heartwood tissues, as reviewed in Refs. [43,179,180], has facilitated the development of wood protection technologies based on the use of natural compounds (bioactives) for improving wood durability [181]. Heartwood formation begins in the centre of the tree stem and is initiated in the heartwood/sapwood interface region, utilizing locally available nutrients as well as nutrients transported via rays across the sapwood. The extractives (non-structural components of wood tissues) produced in parenchyma cells in this interface region are deposited in the cell lumen and often also impregnate the cell wall. The amount, chemistry, and micro-distribution of extractives all deter microbial invasion into wood and influence the natural durability of timbers destined for utilization. It is not therefore surprising that the timbers from trees growing in the tropics, which produce massive volumes of heartwood, are preferred in use where durability is the primary consideration.

From the perspective of the longevity of waterlogged archaeological wooden objects, heartwood no doubt makes important contributions. In heartwood, extractives deposit in the cell lumen and impregnate the cell wall as well as pit membranes, as demonstrated by microscopy tools and techniques, using TEM alone or in combination with LM [182–184]. Some examples of TEM studies in support of the important role that extractives play in the durability of heartwood [185] are warranted. Belian (*Eusideroxylon zwageri*) is a highly sought-after tropical wood species for use, particularly in hazardous environments, because its heartwood is one of the most highly durable timbers in the world. A TEM study to investigate the mode and extent of cell wall degradation in belian heartwood by tunnelling bacteria, the microorganisms regarded as the ultimate degraders, demonstrated that the extremely slow cell wall degradation was correlated with both the amount and distribution of the extractives [182]. In another TEM study, a utility pole, made from the heartwood of the tropical tree species chengal (*Neobalanocarpus heimii*) and inserted into the ground, showed considerable resistance to soft type I attack after years of service [184]. The high degree of resistance was correlated to both the amount and distribution of extractives. Images obtained using LM suggested the almost complete filling of parenchyma and fibre lumens by extractives. TEM images confirmed this and, in addition, provided evidence of extractives

impregnating the cell walls and the pit membranes connecting parenchyma [184], with the resistance to fungal colonization and attack on the cell wall attributable to both physical and chemical constraints. The extractive content of the heartwood of this wood species was high, as also demonstrated in other studies [185].

An appreciation of the importance of heartwood extractives in the longevity of waterlogged heritage woods is also gained from several TEM studies of wooden objects uncovered from extended exposure to burial and waterlogging environments [29,154]. For example, in a study to investigate the microbial degradation of *Pinus sylvestris* wood tissues from a bulwark constructed in early 1100 AD in a lake in Sweden [154], TEM provided evidence that the wood structures that were coated with an extractive layer (such as the warty layer) or were heavily impregnated with extractives (such as the pit membranes between ray tracheids and between axial and ray tracheids) were protected from attacks by erosion bacteria. Kim and Singh [29], examining the state of the microbial degradation of waterlogged archaeological woods from sunken ships using TEM, provided evidence of the resistance of extractive-impregnated pit membranes to erosion bacteria. As the warty layer has to be breached for erosion bacteria to gain entry into the secondary cell wall, and erosion bacteria colonize wood tissues mainly by removing pit membranes, the extractive-mediated protection of these structures would have no doubt slowed down bacterial spread and cell wall degradation, contributing to the longevity of these treasured ancient artefacts.

## 7. Concluding Remarks

The present review focuses on the pivotal role that microscopy has played in understanding the nature of the microbial deterioration of historically and culturally important wooden objects subjected to prolonged burial and waterlogging. Three different microbial degradation patterns have been identified in waterlogged woods [4,6,17]. While the presence of soft rot (Type I) attack was readily diagnosed using LM, the high resolution of electron microscopy, particularly TEM, was needed to unravel the intricacies of bacterial decay patterns (bacterial tunnelling and bacterial erosion).

LM was the main imaging tool for studies undertaken in the 1950s and 1960s to investigate the state and cause of the deterioration of wooden poles (foundation piles) supporting historic buildings. The images captured showed a pattern of wood degradation that did not match any of the familiar patterns produced during cell wall degradation by fungi. Although bacteria were implicated in the decay process, primarily based on their presence in the decaying wood and the recognition that the degradation patterns were very different from fungal decay patterns, this view met with wide scepticism, as all attempts to produce cell wall degradation using single bacterial isolates were unsuccessful, as reviewed in ref. [23]. The TEM examination of ultrathin sections from wood exposed to natural environments confirmed that certain bacteria can indeed degrade wood in its native state, as it became possible to combine the high resolution of TEM with the excellent definition of the features being examined due to the extreme thinness of sections (ultrathin sections). The two distinctly different, intricate patterns produced during the degradation of cell walls were named 'tunnelling' (carried out by tunnelling bacteria) and 'erosion' (carried out by erosion bacteria). The ability to unambiguously diagnose the decay pattern(s) present in waterlogged archaeological woods generated much interest among wood scientists and archaeologists from the perspective of gaining insights into the degradation processes as well as effectively conserving the precious wooden objects uncovered following prolonged burial and waterlogging. Reports from various parts of the world came pouring in, relating exposure conditions to the types of microbial degradation present.

In the ancient waterlogged wooden objects uncovered from the environments supporting the presence of oxygen (oxygenated conditions), such as in shallow waters [142], coastal sites, and intertidal zones [9,32], the presence of soft rot, bacterial tunnelling, and bacterial erosion in differing combinations was reported. This is consistent with the view that soft-rot fungi and tunnelling bacteria require the presence of adequate oxygen, and ero-

sion bacteria can be functional under differing levels of oxygen, including near-anaerobic conditions [19,28]. It became apparent from the many studies undertaken on buried and waterlogged archaeological woods, following the electron microscopic elucidation of bacterial decay patterns, that, under extreme oxygen-limiting conditions, bacterial erosion is the predominant form of cell wall degradation present, such as under deep burial in terrestrial environments [4,6,19,26,28,145] and burial in the ocean floor sediments [4,6]. It is rewarding that bacterial erosion can now be diagnosed using LM alone, based on the knowledge available from combined LM and TEM studies, e.g., Ref. [9], which enables large volumes of ancient woods exposed to prolonged anoxic environments to be rapidly scanned/assessed for microbial deterioration without the need for the tedious and time-consuming preparation and imaging processes associated with electron microscopy. Conservators can now take advantage of the information made available on both the physical state and chemical characteristics of heritage wood at tissue and cell wall levels [5,9,138,186] to develop a more targeted approach for improving the quality of preservation of precious heritage wooden objects as they continue to be uncovered, using natural or bio-based compounds [187–189] compatible with cell walls and cell wall residues where possible.

**Author Contributions:** A.P.S. prepared the original draft and wrote the manuscript. J.S.K., R.R.C. and Y.S.K. contributed to the readability of the manuscript and organized the figures. R.M. contributed to improving the presentation of the manuscript. All authors have read and agreed to the published version of the manuscript.

**Funding:** This research received no external funding.

**Data Availability Statement:** The original contributions presented in the study are included in the article, further inquiries can be directed to the corresponding author.

**Acknowledgments:** The authors would like to thank Charlotte Björdal and Uwe Schmitt for their SEM and TEM images of Figures 10 and 16, respectively. We are also grateful to the journals *Holz als Roh- und Werkstoff*, *IAWA*, and *Forests* for their permission to use the published micrographs in our review paper.

**Conflicts of Interest:** Author Ralf Möller was employed by the company Wolman Wood and Fire Protection GmbH. The remaining authors declare that the research was conducted in the absence of any commercial or financial relationships that could be construed as a potential conflict of interest.

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
