# Peer review of "The Pivotal Role of Microscopy in Unravelling the Nature of Microbial Deterioration of Waterlogged Wood: A Review"

_forests, doi:10.3390/f15050889_

Round 1

Reviewer 1 Report

Comments and Suggestions for Authors

The paper under review presents a valuable overview of the application of various microscopy techniques in analyzing the characteristics of wood, particularly focusing on its structure, chemical composition, and the implications these have for understanding wood's behavior and properties. However, while the paper effectively illustrates the utility of microscopy in wood research, the strength of its arguments varies, and there are areas in which the discussion could be significantly improved or clarified.

The strongest aspect of the paper is its comprehensive synthesis of how different microscopy techniques—confocal, electron, and fluorescence microscopy—are employed in wood science research. For example, the citation of Donaldson et al. (2017) to demonstrate the colocalization of cellulase, lignin, and cellulose is an effective use of specific research to support the argument that fluorescence microscopy is a powerful tool for understanding the interactions between different wood components at the microscopic level.

However, the paper's reliance on some dated references to support its claims is a notable weakness. The field of microscopy and its application in material sciences, including wood research, is rapidly advancing. Technologies that were cutting-edge a decade ago may have been superseded by newer methodologies or found to have limitations that were not initially apparent. As such, the inclusion of more recent studies or a discussion of the evolution of microscopy techniques in wood research would both strengthen the paper's arguments and provide a more accurate picture of the current state of the field.

Additionally, the paper does not critically engage with the limitations or challenges associated with using microscopy in wood research. For instance, the microscopy techniques mentioned are highly sophisticated and require extensive knowledge to interpret the data they produce correctly. There is also the issue of sample preparation, which can significantly affect the results of microscopy studies; this is especially pertinent in the context of wood, where the heterogeneity of the material can lead to variable results. A discussion of these aspects would have provided a more balanced view of the application of microscopy in wood science.

Moreover, the paper's discussion of the Gelbrich et al. (2008) study is a case where the argument could benefit from further validation. The claim that chemical changes in wood degraded by bacteria can be detected using microscopy, based on a limited sample size, presents an opportunity for the original paper to discuss the need for larger-scale studies or alternative methodologies that could confirm or refute these preliminary findings.

In conclusion, while the paper successfully highlights the pivotal role of microscopy in advancing our understanding of wood's micromorphological and chemical properties, its arguments would be significantly strengthened by the inclusion of more current references, a critical evaluation of the challenges associated with the techniques discussed, and a more rigorous examination of the validity of the studies it cites. Addressing these areas would enhance the paper's contribution to the field of wood science and microscopy.

Reviewer 2 Report

Comments and Suggestions for Authors

General opinion:

-          It is a very detailed and in-depth article. It is well-structured and provides clear explanations throughout. Despite its length, I found myself wishing for a few more illustrations depicting some of the anatomical features and pathologies discussed. However, I don't consider this to be a mandatory correction, so to speak.

Comments on the Quality of English Language

-

Reviewer 3 Report

Comments and Suggestions for Authors

This paper present a very detailed review on the structure and ultrastructure of waterlogged wood decayed by microorganisms. The structure and language is sound. However, since the review put emphasis on the role of microscopy in unravelling the nature of microbial deterioration of waterlogged wood, I as well as many readers would expect more information on microscopy. For example, what is the difference among LM, SEM and TEM in terms of sample selection, preparation, result acquisition. What kind of histochemical staining techniques can be applied in LM, SEM, and TEM and their features and applications in interpreting deterioration of waterlogged wood. Therefore, I suggests to include these aspects and simplify 'Composition and Structure of Wood' part that can be found in any book of wood science. Also, adding  several schematic diagrams concerning microscopy and deterioration would be very helpful to summarize the key information of the review and improve the manuscript.

Author Response

  1. Comments/suggestions

Readers would expect more information. For example, differences among LM, SEM and TEM in terms of sample selection, preparation, results, acquisition. What kind of histochemical staining techniques can be applied in LM, SEM, and TEM and their features and applications in interpreting deterioration of waterlogged wood.

Response

In response to the above comments and suggestions and similar suggestions made by the reviewer 2, we have produced a table, providing comprehensive information to address all comments/suggestions (see under reviewer 2).

  1. Comments/suggestions

Simplify 'Composition and Structure of Wood' part that can be found in any book of wood science.

Response

As suggested, 'Composition and Structure' section has been simplified by making changes to the text, including deletion of information that can be readily accessed from books and reviews.

  1. Comments/suggestions

Add several schematic diagrams concerning microscopy and deterioration that can summarise the key information of the review.

Response

A diagram has been added, showing the microbial degradation patterns reported to be present in waterlogged archaeological woods and indicating the type(s) of microscopy that can be used alone and in combination to image patterns important for understanding the decay process as well as for diagnosing the type(s) of degradation patterns. Thus, the diagram presented serves as a composite of several diagrams that can be prepared for the synthesis of above information.

Reviewer 4 Report

Comments and Suggestions for Authors

This is a very comprehensive and informative review. My comments mainly relate to minor edits as the scientific content is excellent.

One query on the AFM, from my experience any roughness in the sample preparation can lead to poor results. Sample preparation is of utmost importance. Could this mean that applying AFM to waterlogged wood/decayed wood is difficult? SHould it be mentioned as a potential constraint?

The paper is quite long, so I would suggest adding a separate section, splitting section 3 at line 668, to create a new section 4 about the Decay types, or Microscopic techniques relevant to each decay type (or some similar title), as the text moves on at this point to discuss specific types of decay and the microscopy relating to them, whereas the earlier paragraphs have outlined the key types of microscopy - Section 3 title could be altered, e.g. Microscopic Techniques. This split might help readers navigate more easily through the large paper.

It is worth also considering opportunities to reduce repetition to reduce the length slightly. One examples is  the sentence on line 367-369, where you don't need to re-state that lignin is 'a polymeric structure that results from polymerisation of different monolignols through oxidative coupling' as this has been said shortly before.

Another potential removal could be on line 500-502. There are two sentences here; the first says the same as the second but the second one contains % values - better to delete the first and keep the second.

Suggested small edits:

line 37 add an 's' to metres as you are saying buried 10 metres deep

Line 42 add 'the' - identify the wood species

Line 157 'cables' seems an unusual word for the cellulose, do you mean crystals? or would a different word work better?

Line 161-2, could remove 'to cell walls' as the sentence would be clearer without this

Line 234 would cellulosic be better as 'cellulolytic' or some other word to indicate cellulse degrading microorganisms?

Line 400 - it could be useful to say early in this paragraph that you are refering to plants such as some palms and bamboos (i.e. not the main 'wood'/ timber species)

Line 595 Better to say 'Another' than 'Other', and to say 'access' than 'accessibility'

Line 625 - typo for 'plant' tissues

Line 738-9, the claues 'particularly common when extensively degraded' would be better in brackets to let the main theme of the sentence - that good preparation leads to sharp images - come over more clearly. Withouth the bracket it could be misleading/confusing.

Line 799 Add 'the' after Although

Line 805 add 'used' after undertaken, and on line 806 add 'that' after bacteria.

Line 807 'could not' might be better than 'cannot' as people were believing it could not, it was not a fact.

Line 816-7, as we are well into 21st Century it could be useful to put 1970s and 1980s, not assume people know you mean that century

Could add 'the' to this sentence too: the late 1970s and throughout the 1980s

Line 1003 could add 'staining' after KMnO4

Line 1063, title could include 'in Compression wood' to be clearer

Line 1087 space needed after comma before TEM

Line 1253 could delete 'presented'

Line 1260 could use 1950s and 1960s, same reason as earlier comment.

Ref 18, there is a typo in the 300,00 year (says 3,000,000 which would be incredible!)

Ref 34 typo for discovered

Also, be careful with the location of spaces between full stops and start of the next sentence. I saw lots of examples so it would be worth checking throughout, but examples for you in line 711, 713, 721

Comments on the Quality of English Language

This is a very well written paper - typos are small and have been noted in the earlier section

Author Response

  1. Query on the AFM

From my experience any roughness in the sample preparation can lead to poor results. Sample preparation is of utmost importance. Could this mean that applying AFM to waterlogged wood/decayed wood is difficult?

Response

The following sentences have been added under 3. Microscopic Techniques: "The quality of sample surface is another limitation in imaging using AFM, as the surfaces to be examined have to be absolutely smooth. However, for waterlogged wood, block face or ultrathin sections from polymer-embedded tissues can overcome this problem, but possible shrinkage of the cell wall resulting from embedding has to be kept in mind.

  1. Comments/suggestions

The paper is quite long, so I would suggest adding a separate section, splitting section 3 to create a new section 4 about the decay types, or microscopic techniques relevant to each decay type.

Response

As suggested, we have split the original section 3 into '3. Microscopic Techniques and 4. Decay Types'.

  1. Comments/suggestions

Lines 367-369 - You don't need to re-state that 'lignin is a polymeric structure that results from polymerisation of different monolignols though oxidative coupling, as this has been said shortly before.

Response

The sentence is deleted, as recommended, and the next sentence modified 'Lignin is a highly irregular (inhomogeneous) molecule---'.

  1. Comments/suggestions

Lines 500-502 - There are two sentences here; the first says the same as the second but the second one contains % values  - better to delete first and keep the second.

Response

The first sentence deleted and the second sentence retained, as recommended.

  1. Comments/suggestions

Line 37 - Add an 's' to metres as you are saying buried 10 metres deep.

Response

'metre' modified to 'metres', as recommended.

  1. Comments/suggestions

Line 42 - Add 'the' - identify the wood species.

Response

'the' added, as recommended.

  1. Comments/suggestions

Line 157 - 'cable seems an unusual word for the cellulose, do you mean crystals? or would a different word work better?

Response

'cable' word deleted, and 'microfibrils' retained.

  1. Comments/suggestions

Lines 161-162 - Could remove 'to cell walls' as the sentence would be clearer without this.

Response

'to the cell wall' deleted.

  1. Comments/suggestions

Line 234 - Would 'cellulosic' be better as 'cellulolytic' or some other word to indicate cellulose degrading microorganisms?

Response

'cellulosic' changed to 'cellulolytic'

  1. Comments/suggestions

Line 400 - It could be useful to say early in this paragraph that you are referring to plants such as some palms and bamboos (i.e. not the main 'wood'/timber species.

Response

All descriptions of palms, bamboos and other monocots  have been deleted in response to Assistant Editor's suggestion. Multilamellar cell walls are described for wood fibres, and not for bamboo and palm fibres.

  1. Comments/suggestions

Line 595 - Better to say 'Another' than 'Other', and to say 'access' than 'accessibility'.

Response

'Other' changed to 'Another' , and 'accessibility' changed to 'access'.

  1. Comments/suggestions

Line 625 - Typo for 'plant' tissues.

Response

Correction made: 'plat' changed to 'plant'.

  1. Comments/suggestions

Lines 738-739 - The clause 'particularly common when extensively degraded' would be better in brackets to let the main theme of the sentence - that good preparation leads to sharp images - come over more clearly.

Response

'(particularly common when extensively degraded)' placed in brackets, as suggested.

  1. Comments/suggestions

Line 799 - Add 'the' after Although.

Response

'the' added after Although.

  1. Comments/suggestions

Line 805 - Add 'used' after undertaken, and on line 806 'that' after bacteria.

Response

'used' added after undertaken.

 But adding 'that' after bacteria will render the sentence grammatically incorrect; changed to 'pure isolates of bacteria'.

  1. Comments/suggestions

Line 807 - 'could not' might be better than 'cannot' as people were believing it could not; it was not a fact.

Response

'cannot' changed to 'could not', as suggested.

  1. Comments/suggestions

Lines 816-817 - As we are well into 21st Century it could be useful to put 1970s and 1980s not assume people know you mean that century; could add 'the' to this sentence too: the late 1970s and throughout the 1980s.

Response

'70s and 80s' changed to 'the late 1970s and throughout the 1980s', as suggested.

  1. Comments/suggestions

Line 1003 - Could add 'staining' after KmnO4.

Response

'staining' added after KMnO4, as suggested.

  1. Comments/suggestions

Line 1063 - Title could include 'in Compression Wood' to be clearer.

Response

'in Compression Wood' added to the title, as suggested.

  1. Comments/suggestions

Line 1087 - Space needed after comma before TEM.

Response

Space added, as suggested.

  1. Comments/suggestions

Line 1253 - Could delete 'presented'.

Response

'presented' deleted, as suggested.

  1. Comments/suggestions

Line 1260 - Could use 1950s and 1960s, same reason as earlier comment.

Response

'50s' and '60s' changed to '1950s and 1960s', as suggested.

  1. Comments/suggestions

Ref 18, there is a typo in the 300, 000 year (says 3,000,000 which would be incredible!).

Response

'3000,000-year-old' corrected to '300,000-year-old'.

  1. Comments/suggestions

Ref 34 typo for discovered.

Response

Spelling corrected.

  1. Comments/suggestions

Also, be careful with the location of spaces between full stops and start of the next sentence. I saw lots of examples so it would be worth checking throughout, but examples for you in lines 711, 713, 721.

Response

We have gone through the entire manuscript and added 'spaces' as recommended, including on lines 711, 713 and 721.

Reviewer 5 Report

Comments and Suggestions for Authors

GENERAL

This review paper was well-written. I enjoyed reading it. It highlights the significant role microscopy has played in understanding microbial deterioration of wooden objects subjected to burial and waterlogging, tracing the evolution of microscopy techniques from light microscopy (LM) to transmission electron microscopy (TEM) and their contributions to identifying different microbial degradation patterns.

The paper discussed how LM and SEM/TEM have been instrumental in diagnosing decay patterns in waterlogged wood, particularly in distinguishing between fungal and bacterial decay. It also emphasizes the importance of TEM in revealing intricate bacterial decay patterns such as tunneling and erosion, which were previously unrecognized using LM alone.

Furthermore, the paper addresses the implications of these findings for conserving waterlogged wooden objects, noting the importance of understanding microbial degradation processes for effective conservation strategies. Importantly, it highlights how advances in microscopy have enabled rapid assessment of microbial deterioration without the need for time-consuming electron microscopy processes.

Overall, this review underscores the pivotal role of microscopy in advancing our understanding of microbial decay in waterlogged wooden objects and its significance for conservation efforts.

I have a few comments that should be incorporated into the revised paper to enhance understanding, especially for readers with limited background in wood anatomy and the macrostructure of softwood and hardwood

2.1 Composition:

Consider tabulating the differences in the proximate chemical composition of hardwoods and softwoods for easy comparison. This will provide readers with a clear and concise reference point to understand the distinct chemical characteristics of each wood type.

2.2 Structure:

To enhance reader comprehension, include illustrations showcasing the macroscopic cellular elements of hardwoods (angiosperms) and softwoods (gymnosperms). Highlighting radial, tangential, and cross-sectional views, as well as distinguishing between sapwood and heartwood, will aid readers in visualizing wood anatomy. Additionally, providing illustrations depicting the major composition differences of cell types between softwoods and hardwoods before delving into discussions on ultrastructure would provide valuable context.

2.3.1 The Typical Three Layered Secondary Wall:

For clarity, consider visualizing the typical arrangement of microfibrils in the three-layered secondary wall. Illustrating a model of microfibril arrangement will serve as a helpful reference for readers, allowing them to better understand the structural organization of the secondary wall.

3. Importance of Microscopy in Evaluating Microbial Decay Types in Ancient Waterlogged Woods:

To facilitate comparison and understanding, consider tabulating the differences and capabilities of each microscope discussed in this section. A tabulated format will allow readers to easily compare the capabilities, uses, and functions of different microscopes, providing a clear overview of their respective contributions to evaluating microbial decay types in ancient waterlogged woods. This visual aid will enhance comprehension and aid readers in selecting appropriate microscopy techniques for their research or conservation efforts.

Author Response

GENERAL

  1. Comments/suggestions

2.1 Composition: Consider tabulating the differences in the proximate chemical composition of hardwoods and softwoods for easy comparison.

Response

The chemical composition of hardwoods and softwoods presented also in a tabular form, as suggested.

  1. Comments/suggestions

2.2 Structure: To enhance reader comprehension, include illustrations showing the macroscopic cellular elements of hardwoods and softwoods. Highlight radial, tangential and cross-sectional views, as well as distinguishing between sapwood and heartwood, will aid reader in visualising wood anatomy.

Response

Reviewer 1 is not supportive of including information/images readily available from wood science books.

  1. Comments/suggestions

2.3.1 The Typical Three Layered Secondary Wall: For clarity, consider visualising the typical arrangement of microfibrils in the three-layered secondary wall, illustrating a model of microfibril arrangement will serve as a helpful reference for reader, allowing them to better understand the structural organisation of the secondary wall.

Response

As suggested, a cell wall model showing microfibril orientation in different cell wall layers is presented.

  1. Comments/suggestions
  2. Importance of Microscopy in Evaluating Microbial Decay Types in Ancient Waterlogged Wood: To facilitate comparison and understanding, consider tabulating the differences and capabilities of each microscope discussed in this section.

Response

As suggested, a comprehensive table, providing comparative information on the differences and capabilities of the commonly used microscopy tools and techniques in assessing ancient waterlogged woods, is presented.